# The *D*-Mercator method for the multidimensional hyperbolic embedding of real networks

Robert Jankowski[1,2], Antoine Allard [3,4], Marián Boguñá [1,2] & M. Ángeles Serrano [1,2,5] ✉

One of the pillars of the geometric approach to networks has been the development of model-based mapping tools that embed real networks in its latent geometry. In particular, the tool Mercator embeds networks into the hyperbolic plane. However, some real networks are better described by the multidimensional formulation of the underlying geometric model. Here, we introduce *D*-Mercator, a model-based embedding method that produces multidimensional maps of real networks into the $(D+1)$-hyperbolic space, where the similarity subspace is represented as a *D*-sphere. We used *D*-Mercator to produce multidimensional hyperbolic maps of real networks and estimated their intrinsic dimensionality in terms of navigability and community structure. Multidimensional representations of real networks are instrumental in the identification of factors that determine connectivity and in elucidating fundamental issues that hinge on dimensionality, such as the presence of universality in critical behavior.

Geometry plays a fundamental role in our understanding of the world and in formulating theories based on geometric principles. In any scientific field, the ability to describe and visualize objects and phenomena with precision is paramount, and geometry serves as a crucial tool for scientific observation, enabling us to perceive, represent, and interpret our surroundings accurately. This transformation of abstract concepts into tangible visualizations facilitates analysis, prediction, and effective communication of scientific ideas. Moreover, measurement lies at the core of scientific inquiry, and geometry provides the framework and necessary tools for precise quantification.

Within this context, the concept of dimension assumes particular relevance as geometric properties associated with measurements on a specific system depend on dimensionality. The physical world we inhabit is three-dimensional, and our understanding is rooted in this framework. However, when confronted with systems or phenomena that exist in higher dimensions, we encounter challenges in making sense of them. In such cases, dimensional reduction becomes necessary. Yet, this process carries the risk of losing or distorting information. Hence, it becomes crucial to carefully choose the appropriate dimension for describing the system. The choice of dimensionality plays a pivotal role in preserving the integrity and accuracy of the information we seek to capture and analyze. By selecting the correct dimension, we can ensure that our descriptions and interpretations remain faithful to the complexities of the system under investigation.

Complex networks are amenable to be described and modeled using geometric postulates. They can be represented in a simplified and comprehensible geometric framework[1] and the dimensionality question can then be addressed from first principles[2]. One might be tempted to think that these principles could be rooted in explicit geometries underlying real systems. For instance, airport networks, urban networks, and power grids connect geographical locations, and 3-dimensional Euclidean space wraps the brain's anatomy. However, in these complex networks, explicit distances explain the tendency of the

[1]Departament de Física de la Matèria Condensada, Universitat de Barcelona, Martí i Franquès 1, 08028 Barcelona, Spain. [2]Universitat de Barcelona Institute of Complex Systems (UBICS), Universitat de Barcelona, Barcelona, Spain. [3]Département de Physique, de Génie Physique et d'optique, Université Laval, Québec, Québec G1V 0A6, Canada. [4]Centre Interdisciplinaire en Modélisation Mathématique, Université Laval, Québec, Québec G1V 0A6, Canada. [5]ICREA, Pg. Lluís Companys 23, E-08010 Barcelona, Spain. ✉e-mail: marian.serrano@ub.edu

elements to be linked to each other only to some extent, and a variety of factors related to structural, functional, and evolutionary constraints are also at play.

A family of simple geometric network models—where distances between nodes on a latent space with hyperbolic geometry integrate the different factors that define the likelihood of interactions[3,4]—has excelled in explaining many fundamental features of real networks. These include the small-world property[5–7], heterogeneous degree distributions[3,4,8], high levels of clustering[4,8–11], self-similarity[3,12,13], and also properties of their spectra such as the spectral gap[14]. These models have been extended to growing networks[15], weighted networks[16], directed networks[17], multilayer networks[18,19], and networks with community structure[20–22].

The discovery of such hidden metric spaces and the understanding of their role have become a major research area leading to network geometry[23] as a new paradigm within network science. In this context, one of the last achievements of hyperbolic network geometry has been the discovery that real networks have ultra-low dimensionality and that networks from different domains show unexpected regularities, including tissue-specific biomolecular networks being extremely low dimensional, brain connectomes being close to the three dimensions of their anatomical embedding, and social networks and the Internet requiring slightly higher dimensionality[2].

Nonetheless, previous embedding tools that map network topologies in its latent hyperbolic geometry assumed that the similarity subspace is one-dimensional[24–33]. Among them, Mercator[33] embeds real networks into the hyperbolic plane on the basis of their congruency with the $\mathbb{S}^1$ model[34], also $\mathbb{H}^2$ in a purely geometric formulation[4], at the core of the network geometry paradigm. The model explains connectivity in real networks by assuming a one-dimensional spherical similarity space plus a popularity dimension that together defines effective distances between nodes in the two-dimensional hyperbolic plane. The likelihood that two nodes are connected decreases with their hyperbolic distance. Mercator uses statistical inference techniques to find the coordinates of the nodes that maximize the congruency between the observed topology and the $\mathbb{S}^1$ model[33,35]. Apart from its accuracy, Mercator has the advantage of systematically inferring not only nodes coordinates but also global model parameters, and has the ability to embed networks with arbitrary degree distributions in reasonable computational time, which makes it competitive for real applications.

Beyond visualization, Mercator maps have been used in a multitude of downstream tasks, including efficient navigation[35–37], the detection of modular organization[21,38], the prediction of missing links[26,39], and the implementation of a renormalization group[13,36,40] that brings to light hidden symmetries in the multiscale nature of real networks and enables scaled-down and scaled-up replicas. Other data-driven techniques have been proposed to embed networks in a latent space where connected nodes are kept close to each other[41–47], but are not comparable as long as distances are not defined in agreement with the relational and connectivity structure of the network, even if the hyperbolic plane was used in some of them as well[48].

The obtained representations are accurate enough in some cases and for certain applications. However, many real complex networks, despite having an ultra-low dimension, are better represented by similarity subspaces of dimension higher than one[2], often with dimensions of $D = 2$ or $D = 3$. Therefore, embeddings in the most suitable dimension for each system have the potential to describe them without the drawbacks of significant dimensional reduction. Here, we introduce $D$-Mercator, a model-based embedding method that leverages two different techniques, model-based Laplacian Eigenmaps (LE) and Maximum Likelihood Estimation (MLE), combining them to produce multidimensional maps of real networks into the $(D+1)$-hyperbolic space, where the similarity subspace is represented as a $D$-dimensional sphere ($D$-sphere). We evaluated the quality of the embedding method using synthetic $\mathbb{S}^D$ networks. We also produced multidimensional hyperbolic maps of real networks. These maps provide more informative descriptions than their two-dimensional counterparts and reproduce the structure of many real networks more faithfully. Multidimensional representations are instrumental in the identification of factors that determine connectivity in real systems and in addressing fundamental issues that hinge on dimensionality, such as the presence of universality in critical behavior. This makes $D$-Mercator a qualitative improvement and not a mere quantitative refinement. $D$-Mercator also allows us to estimate the intrinsic dimensionality of real networks in terms of navigability and community structure, in good agreement with embedding-free estimations[2].

## Results

$D$-Mercator is based on the multidimensional formulation of the geometric soft configuration model, the $\mathbb{S}^D/\mathbb{H}^{D+1}$ model[2,49], which is a multidimensional generalization of the $\mathbb{S}^1$ model[34]. Our approach assumes that real networks are well described by the $\mathbb{S}^D/\mathbb{H}^{D+1}$ model and can be reverse-engineered to infer the coordinates of the nodes and the parameter $\beta$ that give the highest congruency with the observed topology.

In the $\mathbb{S}^D$ model, a node $i$ is assigned a hidden variable representing its popularity, influence, or importance, denoted $\kappa_i$ and named hidden degree. It is also assigned a position in the $D$-dimensional similarity space chosen uniformly at random, and represented as a point on a $D$-dimensional sphere. The $D$-sphere is defined as the set of points in $(D+1)$-dimensional Euclidean space that are situated at a constant distance $R$ from the origin; each node is therefore assigned a vector $\mathbf{v}_i \in \mathbb{R}^{D+1}$ with $\|\mathbf{v}_i\| = R$. The connection probability between a node $i$ and a node $j$ takes the form of a gravity law:

$$p_{ij} = \frac{1}{1 + \chi_{ij}^{\beta}}, \quad \text{with} \quad \chi_{ij} = \frac{R\Delta\theta_{ij}}{\left(\mu \kappa_i \kappa_j\right)^{1/D}}. \tag{1}$$

The number of nodes in the network is $N$ and, for convenience and without loss of generality, we set the density of nodes in the $D$-sphere to one so that

$$R = \left[\frac{N}{2\pi^{\frac{D+1}{2}}}\Gamma\left(\frac{D+1}{2}\right)\right]^{\frac{1}{D}}. \tag{2}$$

The separation $\Delta\theta_{ij} = \arccos(\frac{\mathbf{v}_i \cdot \mathbf{v}_j}{R^2})$ represents the angular distance between nodes $i$ and $j$ in the $D$-dimensional similarity space. The parameter $\beta > D$, named inverse temperature, calibrates the coupling of the network topology with the underlying metric space and controls the level of clustering, which grows with $\beta$ and goes to zero, in the thermodynamic limit, when $\beta \to D^+$. Finally, the parameter $\mu$ controls the average degree of the network and is defined as

$$\mu = \frac{\beta\Gamma\left(\frac{D}{2}\right)\sin\frac{D\pi}{\beta}}{2\pi^{1+\frac{D}{2}}\langle k \rangle}. \tag{3}$$

Hence, given the number of nodes $N$ and the dimensionality $D$ of the similarity subspace, the model is determined by $N(D+1)+1$ parameters: the hidden variables $(\kappa_i, \mathbf{v}_i), i = 1, \dots, N$, and the parameter $\beta$. The hidden degrees can be generated randomly from an arbitrary distribution or taken as a set of prescribed values. The model has the property that the expected value of the degree of a node with hidden variable $\kappa$ is $\bar{k}(\kappa) = \kappa$. An illustration of the $\mathbb{S}^D$ model for $D = 2$ is given in Fig. 1.

The $\mathbb{S}^D$ model can be expressed as a purely geometric model in the hyperbolic space, the $\mathbb{H}^{D+1}$ model[49], by mapping the expected degree of each node $\kappa_i$ to a radial coordinate[50], see details in the

"Methods" section "Proof of the isomorphism between $\mathbb{S}^D$ and $\mathbb{H}^{D+1}$" where we prove the isomorphism between the $\mathbb{S}^D$ and the $\mathbb{H}^{D+1}$ model.

## Multidimensional embedding method

Given a real network with adjacency matrix $\{a_{ij}\}$, the first step in the embedding method requires estimating the nodes' hidden degrees $\kappa_i$ and the inverse temperature $\beta$. This step corrects potential finite-size effects that distort the theoretical correspondence between the expected degree of a node in the $\mathbb{S}^D$ model and its hidden degree. Second, the angular coordinates of nodes are inferred using a model-corrected version of LE. Third, the angular coordinates are refined using MLE. Finally, hidden degrees are readjusted given the newly inferred angular positions.

The estimation of hidden degrees and of the inverse temperature $\beta$ are implemented as an iterative process. The initial value of $\beta$ was chosen randomly between $D$ and $D+1$ so that the model is in the geometric small-world regime. Note, however, that the quality of the inference method does not depend on the value of this initial guess. As the initial values for the hidden degrees $\{\kappa_i, i = 1, ..., N\}$, one can use the observed degrees $\{k_i, i = 1, ..., N\}$ in the original network. The parameter $\mu$ is computed from Eq. (3) using the average of the observed degrees $\langle k \rangle$. The estimation proceeds by adjusting the hidden degrees such that the expected degree of each node in the model matches the observed degree in the original network (see the "Methods" section "Inferring the hidden degrees"). Once the hidden degrees are obtained, the theoretical mean of the local clustering coefficient of networks in the $\mathbb{S}^D$ ensemble can be evaluated (see the "Methods" section "Inferring the inverse temperature $\beta$"). If its value differs from the one of the original network, $\bar{c}$, the value of $\beta$ is adjusted and the process is iterated using the current estimation of hidden degrees until a predetermined precision is reached.

To infer the angular positions of nodes—the vectors $\mathbf{v}_i$—we first find a convenient initial guess using a $\mathbb{S}^D$ model-corrected version of LE. The LE method was originally designed for dimensional reduction

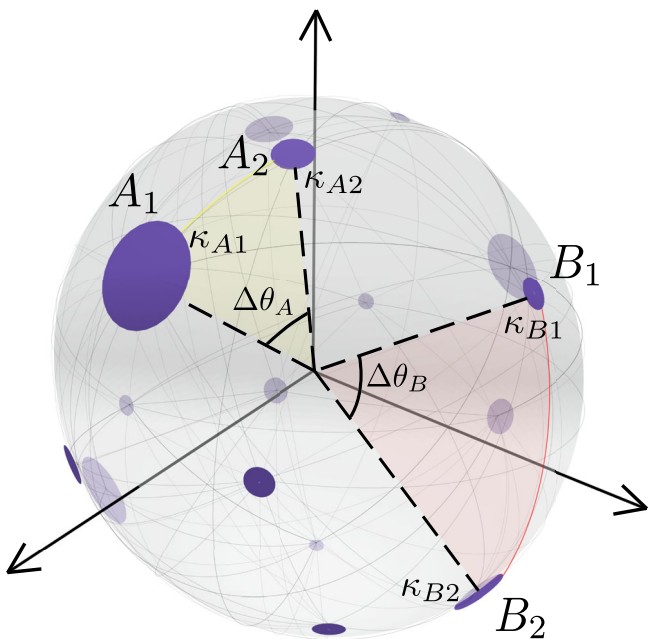

**Fig. 1 | Geometric soft configuration model $\mathbb{S}^D$.** In $D = 2$, the similarity subspace corresponds to the surface of a sphere embedded in three dimensions, such that it can be represented visually. Nodes are placed in the two-dimensional sphere representing the similarity subspace and the size of a node is proportional to its expected degree. The angular distances between pairs of nodes $A_1 - A_2$ and $B_1 - B_2$ are highlighted. Light gray lines on the two-sphere represent connections produced according to the model.

of data in Euclidean space[51] to find the coordinates of points in $\mathbb{R}^m$ given the known distances between pairs of points in $\mathbb{R}^n$, with $m \leq n$. This is achieved by finding a mapping of the set of points $\{\vec{x}_i \in \mathbb{R}^n \to \vec{y}_i \in \mathbb{R}^m\}$ that minimize a given loss function. In $D$-Mercator, the target Euclidean space of the model-corrected version of LE has dimension $D + 1$, and the points to be found $\mathbf{v}_i^{LE}$ define the angular positions of network nodes in $\mathbb{R}^{D+1}$. The loss function is

$$\epsilon_{loss} = \sum_{ij} \left| \mathbf{v}_i^{LE} - \mathbf{v}_j^{LE} \right|^2 \omega_{ij}, \tag{4}$$

where $|\mathbf{v}_i^{LE} - \mathbf{v}_j^{LE}|$ are Euclidean distances between points $i$ and $j$ in $\mathbb{R}^{D+1}$, and the weights $\{\omega_{ij}\}$ are chosen so that the technique can be applied to networks congruent with the $\mathbb{S}^D$ model.

As in standard LE, each weight $\omega_{ij}$ is a decreasing function of the known Euclidean distance between the nodes but, in contrast, only connected pairs contribute to the loss. An approximation to the "known" distances can be inferred from the network structure by using chord lengths in $\mathbb{R}^{D+1}$, so that the weights are set to

$$\omega_{ij} = a_{ij} e^{-\frac{|\mathbf{v}_i - \mathbf{v}_j|^2}{t}} \quad \text{with} \quad |\mathbf{v}_i - \mathbf{v}_j| = 2 \sin \frac{\langle \Delta\theta_{ij} \rangle}{2}, \tag{5}$$

where $\langle \Delta\theta_{ij} \rangle$ is the expected angular distance between nodes $i$ and $j$—with hidden degrees $\kappa_i$ and $\kappa_j$—in the $\mathbb{S}^D$ model and $t$ is a scaling factor fixed as the variance of all the contributing distances. The set of coordinates $\{\mathbf{v}_i^{LE}, i = 1, ..., N\}$ that minimize the loss function above corresponds to the solution of the eigenvalue problem of the weighted Laplacian matrix $L_{ij} = I_{ij} - \omega_{ij}$, where $I$ is the diagonal matrix with entries $I_{ii} = \sum_j \omega_{ij}$, so that $v_j^{LE,i}$ is the $i$-th component of the $j$-th Laplacian eigenvector with non-null eigenvalue (the eigenvectors are ordered according to their eigenvalues). Fortunately, for sparse networks, there exists very fast algorithms able to solve the eigenvalue problem of weighted Laplacians[52], so this step of the method is not computationally expensive. Finally, the positions found by solving the eigenvalue problem are then normalized so that all points lay on the $D$-sphere of radius $R$, i.e., $\mathbf{v}_i = R\mathbf{v}_i^{LE}/(||\mathbf{v}_i^{LE}||)$. Note that, since degree-one nodes do not add geometric information, we remove them from the network and add them back once the coordinates of their neighbors are found (see the "Methods" section " $\mathbb{S}^D$ model-corrected Laplacian Eigenmaps").

Using the coordinates inferred by LE as the initial condition, the coordinates in the similarity subspace are fine-tuned by Maximum Likelihood Estimation (MLE) to optimize the probability for the observed network to be generated by the $\mathbb{S}^D$ model (see the "Methods" section "Likelihood maximization"). Nodes are visited sequentially and new positions are proposed in the vicinity of the mean vector of the node's neighbors. The most favorable proposed position, the one maximizing the local log-likelihood in Eq. (23), is selected and the process is repeated until the local log-likelihood function reaches a plateau. Notice that the final angular coordinates could be improved further by repeating the refining step taking as initial condition the previous MLE inference.

The embedding method ends after the hidden degrees are finally readjusted to compensate deviations from $\bar{k}(\kappa_i) = k_i$, which might have been introduced in the process of estimating the coordinates of nodes in the similarity subspace (see the "Methods" section "Final adjustment of hidden degrees"). An implementation of $D$-Mercator is publicly available at https://github.com/networkgeometry/d-mercator.

The complexity of Mercator is $\mathcal{O}(N^2)$ for sparse networks with $N$ nodes. $D$-Mercator operates on the basis of the $\mathbb{S}^D$ model for an arbitrary value of the dimension $D$, which requires general equations for inferring hidden degrees and parameter $\beta$. This makes some equations impossible to solve analytically, and numerical integrations are

needed. Therefore, the time complexity of the method increases slightly compared to Mercator, but only affects pre-factors and not the scaling with the system size. The detailed computational complexity comparison between Mercator and $D$-Mercator is shown in Supplementary Fig. S2.

### Validation in synthetic networks

**Quality of the embeddings.** The most stringent way to assess whether a map produced by $D$-Mercator is reliable consists of testing synthetic networks generated with the $\mathbb{S}^D$ model with different topological properties and dimensions. The produced networks can then be embedded with the same dimension used to generate them or with a different dimension. In this case, the network's ground-truth is known and the accuracy of the embedding can be evaluated by implementing quality measures of congruency. In Fig. 2, we show an example of the capability of $D$-Mercator to recover the correct coordinates of nodes in synthetic $\mathbb{S}^2$ networks with different values of the exponent of the scale-free power-law degree distribution $\gamma$ and inverse temperature $\beta$. Notice that the agreement between the original coordinates and the inferred ones is excellent, with values for the corresponding Pearson correlation coefficient above 0.96. Results for other values of $\beta$ and $\gamma$ with dimension $D = 3$ are reported in Supplementary Fig. S3 with similar accuracy.

We also checked the inference of the parameter $\beta$ (see Supplementary Fig. S4) as well as the agreement between the empirical and the theoretical connection probabilities (left bottom panels of Supplementary Figs. S5–S14), where the empirical connection probability is measured as the fraction of connected pairs as a function of the rescaled distance $\chi_{ij}$. Again, we found the inference of $\beta$ to be very precise for all the considered synthetic networks, and the theoretical curve for the connection probability is well recovered. Altogether, these results confirm that $D$-Mercator is not just a high-fidelity algorithm in terms of the reconstruction of the similarity coordinates of synthetic networks, but it also correctly determines all other model parameters, including hidden degrees and inverse temperature, and it does so independently of the dimensionality of the network.

Next, we show that $D$-Mercator is able to identify the correct dimension used to generate $\mathbb{S}^D$ synthetic networks without prior knowledge of $D$.

**Navigability.** We studied the navigability of $D$-Mercator maps using greedy routing (GR)[53]. In GR, messages are transferred on the network from a source to a target destination by repeatedly forwarding the message to the neighboring node that is the closest to the target in the multidimensional hyperbolic map. In particular, we investigated whether the native dimension of the network gives the best result when taken as the dimension of the embedding space as compared to other values. Typically, hyperbolic maps of real networks in $D = 1$ display high navigability in the region of high clustering and heterogenous degree distributions[53]. Hence, we generated synthetic networks with a specific dimensionality and these topological characteristics and obtained hyperbolic maps by embedding them using $D$-Mercator with different values of the embedding dimension.

The performance of greedy routing is evaluated by the fraction of successful attempts—when messages reach their destination—, and by their stretch defined as a ratio between the hop-lengths of successful greedy paths and the corresponding shortest paths in the network. In Fig. 3a–d, we show the success rate as a function of the embedding dimension for networks generated in $\mathbb{S}^1$ to $\mathbb{S}^4$. In all cases, the fraction of successful paths varies across $D$. The corresponding stretch values always remain low and vary consistently, although only slightly, with the success rate, such that the highest success corresponds to the lowest stretch (see Supplementary Fig. S16). This means that $D$ does not need to be optimized across the two dimensions of success rate and stretch, but one can only focus on the success rate. Strikingly, the success rate is markedly higher when networks are embedded in their native dimension, meaning that geodesics and shortest paths are the most congruent in the native dimension. For instance, $\mathbb{S}^2$ networks have the highest success when embedded in $D = 2$, for which the lowest stretch is also observed. This implies that the performance of greedy routing, and in particular of the success rate, in maps produced by $D$-Mercator for different values of the embedding dimension $D$ can help identify the intrinsic dimensionality of real networks.

**Geometric community concentration.** Multidimensional hyperbolic maps of networks are especially convenient to explore their community structure[54–56]. In Mercator maps, geometric communities are defined as regions of the similarity subspace densely populated with richly interconnected nodes[20,21]. Real networks in a variety of domains were found to display geometric communities that correlate well with metadata not informed to the algorithm, for instance, world regions in Internet[35] or WTW maps[38], biological pathways in metabolic networks[57], and anatomic brain regions in human connectomes[37].

In synthetic networks, we found that embeddings in dimensions higher than that of the original network are still informative of geometric communities while the opposite is not true, especially when clustering is moderate or low (see Supplementary Fig. S17). The intuitive explanation is that it is always possible to find an isometry

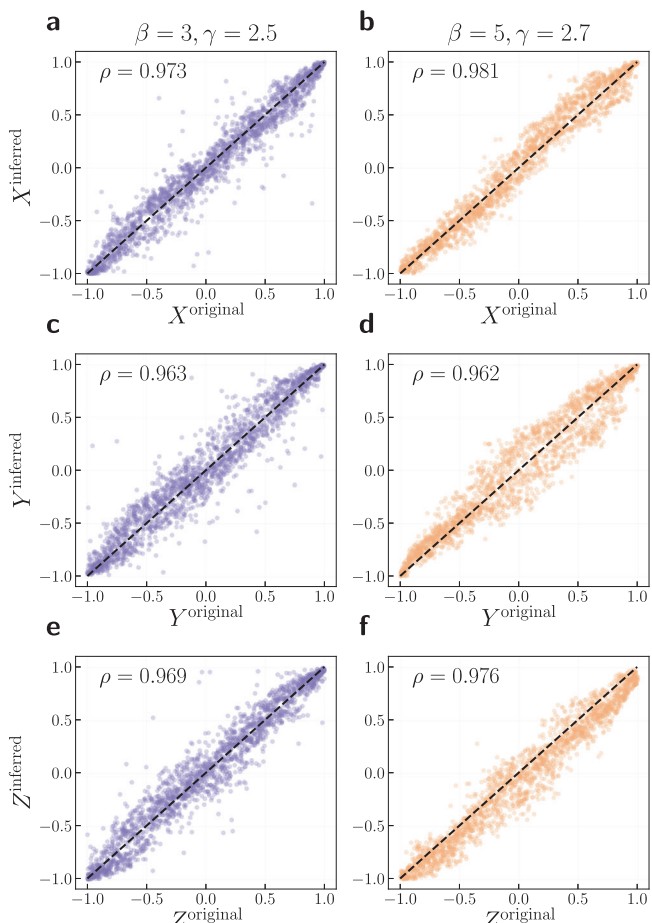

**Fig. 2 | Validation of $D$-Mercator on synthetic networks.** Relationship between coordinates of the synthetic $\mathbb{S}^2$ networks (original) and its embeddings (inferred) with parameters: **a, c, e** $\beta = 3, \gamma = 2.5, N = 2000, \langle k \rangle = 8$ and **b, d, f** $\beta = 5, \gamma = 2.7, N = 2000, \langle k \rangle = 9$. In the top left corner of each figure, the value of the Pearson correlation coefficient between the inferred and original coordinates is reported. Since the coordinates from the embedding might be rotated, we transform them to minimize the average angular distance between the original and inferred coordinates (see Section III in Supplementary Information).

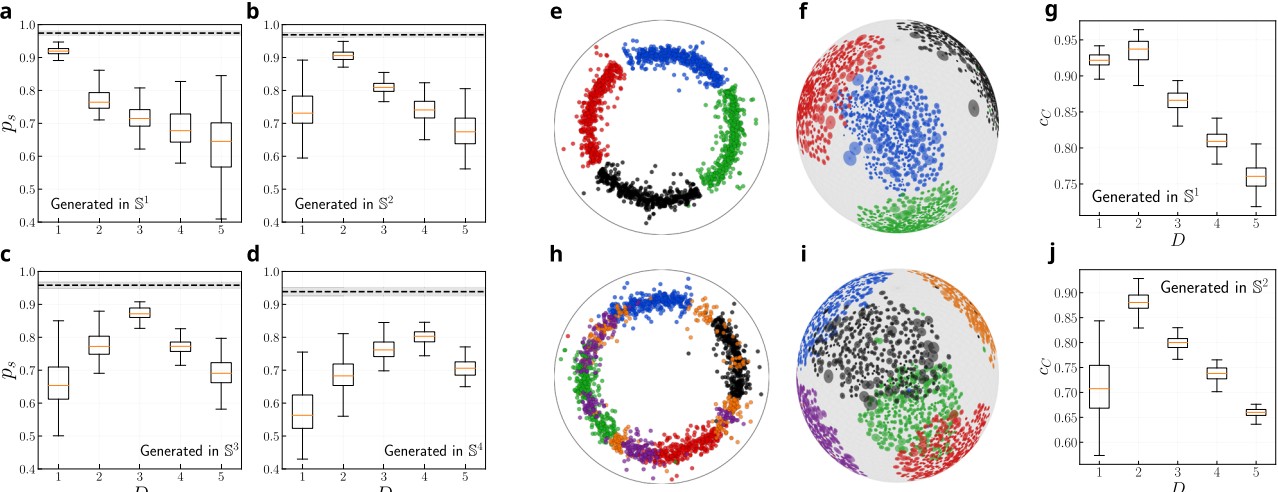

**Fig. 3 | Detecting the dimensionality of synthetic networks. a–d** Greedy routing in multidimensional hyperbolic maps of synthetic networks. The probability of successful paths ($p_s$) as a function of the embedding dimension for $\mathbb{S}^D$ synthetic networks generated in dimensions **a** $D = 1$, **b** $D = 2$, **c** $D = 3$ and **d** $D = 4$. The black dashed lines show the maximum value of $p_s$ for each dimension computed from the generated synthetic networks using the real coordinates. The box ranges from the first quartile to the third quartile. A horizontal line goes through the box at the median. The whiskers go from each quartile to the minimum or maximum. Results obtained by averaging over 100 realizations with $\beta = 2.5D$, $\gamma = 2.7$ and $N = 2000$. **e–j** Community concentration in multidimensional hyperbolic maps of synthetic networks with modular structure. The community concentration $c_C$ for **g** the $\mathbb{S}^1$ model with 4 communities and **j** the $\mathbb{S}^2$ model with 6 communities embedded in different dimensions. Visualization of the embedding of $\mathbb{S}^1$ model with 4 communities in **e** $D = 1$ and **f** $D = 2$. Visualization of the embedding of $\mathbb{S}^2$ model with 6 communities in **h** $D = 1$ and **i** $D = 2$. Nodes are colored based on their communities, and their size in (**f**) and (**i**) is proportional to their expected degree. Results obtained by averaging over 50 realizations with $\beta = 1.5D$, $\gamma = 2.7$ and $N = 2000$.

between a set of points in a metric space and a space of higher dimensionality, whereas the opposite is, in general, not true. Specifically, we generated networks in $\mathbb{S}^1$ with four geometrically localized communities and in $\mathbb{S}^2$ with six geometrically localized communities and obtained *D*-Mercator maps using different embedding dimensions. To generate the synthetic communities, we defined spherical caps with the apices evenly distributed on the surface of the *D*-sphere and polar angle $\Delta\theta_T = 0.7$. Finally, nodes in each community were distributed homogeneously within the corresponding cap. The selected value of $\Delta\theta_T$ organizes nodes into non-overlapping communities. Once the $\Delta\theta_T > \pi/4$, the communities become mixed and overlap increases, as shown in Supplementary Fig. S18.

To measure how the communities remained clustered, we measured the geometric concentration of a community *l* around a node *i* as

$$\rho_{i,l} = \frac{n_{i,l}}{n_{i,g}} \frac{N}{N_l} \qquad (6)$$

where $n_{i,l}$ is the number of nodes in community *l* out of $n_{i,g}$ considered nodes, and $N_l$ is the total number of nodes in the community *l*. The denominator $n_{i,g}$ can be determined in different ways. Here, we take $n_{i,g}$ as the top geometrically closest neighbors. Notice that with this definition, the limit $n_{i,g} \to N$ implies that $\rho_{i,l}$ goes to 1. To obtain the community concentration for a given network map as a single scalar, $\rho$, we restricted the computation to the community to which each node belongs and averaged over all nodes in the network. Notice that $n_{i,g} \to N$ implies that $\rho \to \rho_{ran} = 1/N_C$, where $N_C$ is the number of communities. Finally, we calculated the community concentration as $c_C = \rho(n_{i,g} = N/10)$, i.e., the geometric concentration at 10% of top geometrically closest nodes.

The results reported in Fig. 3g show that embeddings in $D = 1$ and $D = 2$ of the clustered $\mathbb{S}^1$ synthetic networks display similar geometric concentration of communities, and are clearly visually discernible in the maps for both dimensions, as shown in Fig. 3e, f. In contrast, when the clustered networks are produced in $D = 2$, the concentration of communities in one-dimensional maps is clearly worse (see Fig. 3j), with some communities separated in different chunks and scattered all over

the circle, as shown in Fig. 3h. This result illustrates the importance of choosing the most appropriate dimension. Indeed, in this case, the one-dimensional embedding would result in a very inaccurate quantitative description of the system. Despite this result, interestingly, the validation of the topological properties of the graphs shows that embeddings in both dimensions can largely replicate the properties of the network, such as degree distribution, clustering spectrum or average nearest neighbors degree (Supplementary Figs. S20 and S21). These results explain why one-dimensional embeddings have been quite good for many real networks, even if their natural dimension is greater than 1. However, while one-dimensional maps provide information about the community organization of the network, two-dimensional maps offer a much richer and more accurate representation. Finally, when we increase the value of $\Delta\theta_T$, thus introducing the mixing of communities, the findings remain consistent (see Supplementary Fig. S19). Therefore embedding in the higher dimension is needed to uncover the community structure of the networks.

The analysis above shows that there is a strong congruency between ground-truth communities and embeddings obtained by *D*-Mercator when the appropriate dimension is selected, as nodes are surrounded mainly by other nodes in the same community. In turn, this result implies that such embeddings can be used to detect similarity-based communities even when the metadata defining groups is not available.

### Multidimensional maps of real-world networks

We compiled data for several real-world complex networks from different domains and embedded them in different dimensions. More specifically, we generated results for six real networks for which metadata reporting categories are available: a sample of the Add-Health study, where adolescent students in six grades are linked by social interactions[58]; the network of international trade in apples from the Food and Agricultural Organization (FAO) of the United Nations[59] (see Supplementary Fig. S22); the neural network of the *C. elegans* worm, where communities are the different neurons' classes[60] (see Supplementary Fig. S23); the network OpenFlights for flights between airports around the world[61]; Foxglove (*Digitalis purpurea*) network[62]

**Table 1 | Properties of the real networks used in this work**

| Network | N | ⟨k⟩ | c̄ | $N_C$ | β | μ | D |
|---|---|---|---|---|---|---|---|
| Add-health | 1996 | 8.49 | 0.15 | 6 | 2.60 | 0.0104 | 2 |
| FAO-apples | 152 | 15.99 | 0.56 | 6 | 3.98 | 0.0126 | 2 |
| *C. elegans* | 559 | 16.1 | 0.32 | 5 | 3.16 | 0.0091 | 2 |
| OpenFlights | 2905 | 10.77 | 0.59 | 6 | 1.85 | 0.0272 | 1 |
| Foxglove | 2916 | 11.21 | 0.43 | 8 | 10.23 | 0.0184 | 3 |
| Polbooks | 105 | 8.4 | 0.49 | 3 | 4.75 | 0.0278 | 2 |

The β and μ values are presented for the best dimension of the embeddings. The $N_C$ indicates the number of ground-truth communities, while D is the inferred dimension as determined by the performance of greedy routing, community concentration, and community detection.

which describes the global organ-wide cellular connectivity of plant hypocotyls (see Supplementary Fig. S25). The links between cells were identified by detecting common surfaces using 3D cellular meshes that represent the intercellular association; and Polbooks network (see Supplementary Fig. S26)[63] where nodes represent the books about U.S. politics published close to the 2004 U.S. presidential election, and sold by Amazon.com. Edges between books represent frequent copurchasing of those books by the same buyers. The nodes' attributes indicate the political leaning: liberal, conservative or moderate. In the geographical networks, we took continents as categories. We report global statistics of the networks in Table 1 and Supplementary Table S1.

Results validating the embeddings in terms of topological congruency between the network and the inferred model can be found in Supplementary Figs. S27–S32, where we show the degree distribution, the average nearest neighbors degree, the number of triangles, and the clustering distribution. The probability of connection and other local properties are well reproduced in all dimensions. However, as displayed in the bottom panels of Fig. 4, and taking the Add-health network as a case example, not all dimensions provide the same efficiency in terms of GR success rate $p_s$, community concentration $c_C$, and performance of community detection. To detect the communities, we employed the hierarchical clustering algorithm to cluster nodes in the D-dimensional similarity space for the detection of communities. One popular method, called the agglomerative clustering algorithm from the scikit-learn library[64], merges successive nodes close to each other in the similarity space, i.e., when they are separated by a small angular distance. The approach can be applied to real networks with a predefined number of clusters, for instance as given in the metadata. To assess the performance of the community detection algorithm, we measured the modularity of the network[54], Q, and also computed the Normalized Mutual Information (NMI) between the predicted communities and the metadata labels. We compared the geometry-based agglomerative clustering method with several topology-based alternatives (see Section XIII in Supplementary Information). Overall, the geometric community detection method exhibits comparable outcomes regarding modularity and NMI.

For the networks analyzed in this work, the four metrics provide coinciding information and each network has a specific dimension that is clearly better. Hence, we propose that the dimension of a network is the consensus value of D among the analyzed structural features including congruency with metadata. Following this prescription, the hyperbolic dimension of Add-health displayed in Fig. 4 is D + 1 = 3, such that the similarity subspace can be easily visualized as a 2-sphere, as shown in Fig. 4b. Interestingly, performing a one-dimensional embedding results in some of the communities mixed up see Fig. 4a. This is the case of the 7th and 8th grades, which appear completely mixed up in the one-dimensional embedding whereas in the two-dimensional one both grades are well separated. Again, this result clarifies the importance of using the most appropriate dimension for the description of the system.

For OpenFlights, we found that the best hyperbolic dimension is D + 1 = 2, see Supplementary Fig. S24, which indicates that the topology of the airports network is 1-dimensional in the Euclidean similarity space whereas one could have naively expected D = 2. This last result highlights the distinction between the geometry of the Earth and that of the topology of the airports network, where long-range flights between hubs reduce the effective geometry of the planet. The embedding of Foxglove revealed that the greedy routing and community concentration achieve the highest values in D + 1 = 4. This finding is consistent with the fact that the network is a 3D geometric graph of intercellular associations. Furthermore, our analysis revealed that the Polbooks network can be most accurately represented in D + 1 = 3 dimensions. Although the highest community concentrations are found at both D + 1 = 2 and D + 1 = 3, the performance of greedy routing makes it evident that D + 1 = 3 captures better the network topology.

In all the networks analyzed in this work, the community concentration at the optimal dimension is significantly larger than the random case $N_C^{-1}$, thereby validating the quality of embeddings found by D-Mercator.

The similarity maps in the 2-sphere show interesting information about the spatial distribution of communities and their relation with categories as defined by metadata. In the Add-Health network, there are six categories corresponding to the grades the students belong to. One can observe that students in lower grades (classes 7 and 8) are clearly separable in the D = 2 similarity map and are mixed in the lower dimension. In contrast, nodes' positions of adolescents in the 10th to 12th grades are mixed, indicating that friendships were formed between members of the different classes. The countries in the FAO-Apples embedding are quite clearly grouped into continents. The European countries are placed in one similarity region, whereas nodes from Asia and Oceania are positioned on the opposite side of the sphere, thereby outlining the interconnectivity of trades of apples within Europe and largely not between different continents. Finally, the neurons in *C. elegans* are divided into five categories including motor, sensor, interneurons, neurons in the pharynx, and sex-specific neurons. Again, the different categories are clearly separated.

## Discussion

Throughout history, maps have been at the center of political, economic, and geostrategic decisions to become a critical piece in our everyday lives, serving as an integral, accurate, and relevant information source. Their appeal is not only visual, they provide a way of storing and presenting information and communicating findings, they let us recognize locational distributions and spatial patterns and relationships, and they make it possible for us to conceptualize processes that operate through space. Our overarching goal is to map real-world complex systems in an embedding metric space that ought not to be geographical or spatially obvious, but that may be a condensate of the different intrinsic attributes that determine how distant, conversely similar, the elements of the system are.

Maps in the hyperbolic plane obtained by $\mathbb{S}^1$ model-based optimization are meaningful representations that explain the observed regularities in the interaction fabric of real networks and have been used in a multitude of downstream tasks. In some cases, networks are intrinsically one-dimensional, and, in general, D = 1 maps offer a good approximation. But, in most cases, multidimensional hyperbolic embeddings of real networks with D > 1 provide more accurate descriptions and will help to discern the role of the different attributes that determine the connectivity in complex systems—such as, for instance, the specific role of geographic and cultural factors in economic and social networks.

Community detection will also benefit from multidimensional hyperbolic embeddings, which facilitate the application of the large

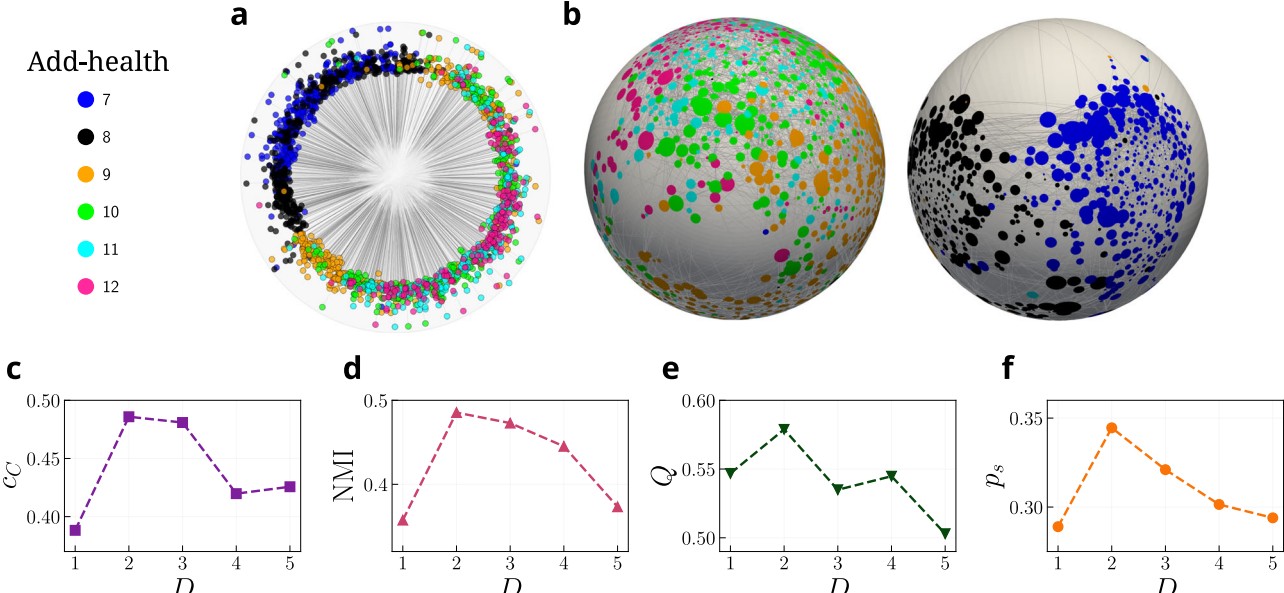

**Fig. 4 | Case study: the Add-health dataset.** Top panels show the embeddings in **a** $D = 1$ and **b** two perspectives of the $D = 2$ similarity space of $D$-Mercator embeddings of the network. The size of a node is proportional to its expected degree, and its color indicates the community it belongs to. For the sake of clarity, only the connections with probability $p_{ij} > 0.5$ given by Eq. (1) are shown. Bottom panels show the performance of **c** community concentration ($c_C$), **d** community detection (NMI), **e** modularity ($Q$), and **f** the success rate of GR ($p_s$).

family of methods based on geometric information and spatial distances[65–67]. We found here that the embedding of a real network in the proper dimension yields the partition of nodes with the highest modularity using a hierarchical clustering algorithm in the similarity subspace. However, the interplay of dimensionality with geometric community detection algorithms is not obvious and this issue will require future investigations. In relation to this, it is also worth mentioning that the interplay between the hyperbolic community structure in higher dimensions and the performance of a variety of tasks is not fully understood yet[50]. A further step would be to scrutinize the effect of the coupling between the dimensionality and performance of greedy routing when the conformation of communities is being changed.

From a technical point of view, the problem of producing geometric network embeddings is NP hard, as the majority of optimization and network inference problems, and the solution can be trapped in some local optima. This is what makes particularly important a proper validation protocol on the basis of synthetic networks produced by the model such that the ground truth is known and the results can be compared against it. We believe that machine learning techniques will provide complementary tools in the future for inferring nodes' coordinates and model parameters that can more accurately approach the global optimum. This will maximize even more the likelihood of the hyperbolic model reproducing the original network. This approach could also be helpful for the optimization of the algorithm to deal with larger networks.

Other lines of research for future work refer to the extension of $D$-Mercator to embed networks with weak geometric structure[11]. Applications such as geometric renormalization[12], link prediction[26], community detection[55,56], studying geometric temporal networks[68] and bipartite networks[69], and the analysis of geometric Turing patterns[70], will definitely benefit from the representation of complex networks in their optimal dimension. Beyond network science, our low dimensional representation can impact fields like machine learning in the short term, where it can be used to improve the relational structure that determines the aggregation and message-passing steps of graph neural networks[71].

## Methods
### Proof of the isomorphism between $\mathbb{S}^D$ and $\mathbb{H}^{D+1}$
The $\mathbb{S}^D$ model can be expressed as a purely geometric model in the hyperbolic space, the $\mathbb{H}^{D+1}$ model[49], by mapping the expected degree of each node $\kappa_i$ to a radial coordinate as[50]

$$r_i = \hat{R} - \frac{2}{D} \ln \frac{\kappa_i}{\kappa_0}, \text{ with } \hat{R} = 2 \ln \left( \frac{2R}{(\mu \kappa_0^2)^{1/D}} \right). \quad (7)$$

Using this transformation, the connection probability of the $\mathbb{S}^D$ model given in Eq. (1) can be rewritten in the form

$$p_{ij} = \frac{1}{1 + e^{\beta[\ln(R\Delta\theta_{ij}) - \frac{1}{D}\ln(\mu\kappa_i\kappa_j)]}} \equiv \frac{1}{1 + e^A}. \quad (8)$$

Hidden degrees $\kappa_i$ and $\kappa_j$ can then be transformed into radial coordinates using Eq. (7), such that $\kappa_i = \kappa_0 e^{\frac{D}{2}(\hat{R} - r_i)}$. Skipping intermediate steps, the term $A$ is

$$A = \beta[\ln(R\Delta\theta_{ij}) - \frac{1}{D}\ln(\mu\kappa_i\kappa_j)] \quad (9)$$

$$= \frac{\beta}{2}[r_i + r_j + 2\ln\left(\frac{\Delta\theta_{ij}}{2}\right) - \hat{R}] \quad (10)$$

$$= \frac{\beta}{2}[x_{ij} - \hat{R}]. \quad (11)$$

Thus, we finally obtain

$$p_{ij} = \frac{1}{1 + e^{\frac{\beta}{2}(x_{ij} - \hat{R})}}, \text{ with } x_{ij} = r_i + r_j + 2\ln\frac{\Delta\theta_{ij}}{2}, \quad (12)$$

which is the connection probability in the $\mathbb{H}^{D+1}$ model. The quantity $x_{ij}$ is a good approximation of the hyperbolic distance between two nodes with radial coordinates $r_i$ and $r_j$ separated by an angular distance $\Delta\theta_{ij}$. This approximation is very accurate for pairs of nodes separated by

$\Delta\theta_{ij} \gg 2\sqrt{e^{-2r_i} + e^{-2r_j}}$[4], the fraction of which converges to one in the thermodynamic limit[1].

## D-Mercator in details

**Inferring the hidden degrees.** Given a value of $\beta$ and the corresponding value of $\mu$ from Eq. (3),

1. *Initialize the hidden degrees* by setting $\kappa_i = k_i$, $\forall_{i=1,\ldots,N}$, where $k_i$ is the observed degree of node $i$ in the real network.
2. *Compute the expected degree for each node $i$* according to the $\mathbb{S}^D$ model as

$$\bar{k}(\kappa_i) = \frac{\Gamma\left(\frac{D+1}{2}\right)}{\sqrt{\pi}\Gamma\left(\frac{D}{2}\right)} \sum_{j \neq i} \int_0^\pi \frac{\sin^{D-1}\theta \, d\theta}{1 + \left(\frac{R\theta}{(\mu\kappa_i\kappa_j)^{1/D}}\right)^\beta}. \tag{13}$$

3. *Adjust the hidden degrees.* Let $\epsilon_{\max} = \max_i\{|\bar{k}(\kappa_i) - k_i|\}$ be the maximal difference between the actual degrees and the expected degrees, and $\epsilon$ a tolerance parameter.

   If $\epsilon_{\max} > \epsilon$, the set of hidden degrees needs to be corrected. To do so, we set $|\kappa_i + [k_i - \bar{k}(\kappa_i)]u| \to \kappa_i$ for every class of degree $k_i$, where $u \cdot U(0,1)$. The random variable $u$ prevents the process from getting trapped in a local minimum. Next, go to step 2 to compute the expected degrees corresponding to the new set of hidden degrees.

   Otherwise, if $\epsilon_{\max} \leq \epsilon$, the hidden degrees have been correctly inferred for the current global parameters. The tolerance parameter used in this work was set to $\epsilon = 0.01$.

**Inferring the inverse temperature $\beta$.** Inferring $\beta$ requires computing the expected local mean clustering $\bar{c}$, given the current values of the global parameters as well as the hidden degrees $\kappa(k)$ computed in the "Methods" section "Inferring the hidden degrees".

The method D-Mercator uses is based on the following. Suppose that we want to estimate the expected clustering $\bar{c}$ of some node of degree $k$. According to the definition of mean local clustering, this quantity is the probability for two randomly chosen neighbors of the node to be connected, which can be computed by following these two steps. First, we randomly choose two of its neighbors and draw their distances to the node from the distribution of distances between connected nodes in the model. Second, we compute the distance between the two neighbors and, with it, the probability for them to be connected.

Two important points require further clarification.

- The model is uncorrelated at the hidden level. Thus, we can draw two neighbors from the uncorrelated distribution $P(k|k') = kP(k)/\langle k \rangle$.
- The distribution of angular distance $\Delta\theta$ between two connected nodes with hidden degrees $\kappa$ and $\kappa'$ reads

$$\rho(\Delta\theta|a_{\kappa\kappa'} = 1) = \frac{p(a_{\kappa\kappa'} = 1|\Delta\theta)\rho(\Delta\theta)}{p(a_{\kappa\kappa'} = 1)}, \tag{14}$$

where $p(a_{\kappa\kappa'} = 1|\Delta\theta)$ is the probability that two nodes with hidden degrees $\kappa$ and $\kappa'$ separated by a distance $\Delta\theta$ and in $D$-dimensional space are connected given by Eq. (1). This probability is

$$p(a_{\kappa\kappa'} = 1|\Delta\theta) = \frac{1}{1 + \left(\frac{R\Delta\theta}{(\mu\kappa'\kappa)^{1/D}}\right)^\beta}. \tag{15}$$

The distribution of distances in the $\mathbb{S}^D$ model is

$$\rho(\Delta\theta) = \frac{\Gamma\left(\frac{D+1}{2}\right)\sin^{D-1}\Delta\theta}{\Gamma\left(\frac{D}{2}\right)\sqrt{\pi}} \tag{16}$$

which becomes increasingly peaked around $\Delta\theta = \pi/2$ as $D \to \infty$. Finally, $p(a_{\kappa\kappa'} = 1)$ is the connection probability between two nodes with hidden degrees $\kappa$ and $\kappa'$ and is given by

$$p(a_{\kappa\kappa'} = 1) = \int_0^\pi \frac{\sin^{D-1}\Delta\theta \, d\Delta\theta}{1 + \left(\frac{R\Delta\theta}{(\mu\kappa'\kappa)^{1/D}}\right)^\beta} \tag{17}$$

Equation (14) therefore reads

$$\rho(\Delta\theta|a_{\kappa\kappa'} = 1) = \frac{\frac{\sin^{D-1}\Delta\theta}{1 + \left(\frac{R\Delta\theta}{(\mu\kappa'\kappa)^{1/D}}\right)^\beta}}{\int_0^\pi \frac{\sin^{D-1}\Delta\theta d\Delta\theta}{1 + \left(\frac{R\Delta\theta}{(\mu\kappa'\kappa)^{1/D}}\right)^\beta}} \tag{18}$$

With these tools in hand, the expected mean clustering is estimated as follows.

1. *Initialize mean local clustering*: Let $\bar{c}(k)$ represent the expected mean local clustering of degree class $k$. Set $\bar{c}(k) = 0$ for all $k$.
2. *Compute the expected mean local clustering spectrum*: For every degree class $k$, repeat $m$ times:
   (a) Draw two variables $k_i$ from $P(k_i|k)$, $i = 1, 2$.
   (b) Draw the corresponding random variable $\Delta\theta_i$ from the distribution $\rho(\Delta\theta_i|a_{\kappa(k)\kappa(k_i)} = 1)$, $i = 1, 2$ given in Eq. (14).
   (c) Generate two random vectors $\mathbf{v}_i$, $i = 1, 2$ with a given angular separation $\Delta\theta_i$, $i = 1, 2$ and compute the angular distance: $\Delta\theta_{12} = \arccos(\mathbf{v}_1 \cdot \mathbf{v}_2)$.
   (d) Set $\bar{c}(k) + p_{12}/m \to \bar{c}(k)$, where $p_{12} = \frac{1}{1 + \left(\frac{R\Delta\theta_{12}}{(\mu\kappa(k_1)\kappa(k_2))^{1/D}}\right)^\beta}$ is a probability for nodes 1 and 2 to be connected.
3. *Compute the expected mean local clustering* as $\bar{c} = \sum_k \bar{c}(k)N_k/N$.

If the error in the expected mean local clustering is $|\bar{c} - \bar{c}^{\text{emp}}| < \epsilon_{\bar{c}}$, where $\bar{c}^{\text{emp}}$ is the mean local clustering of the network to be embedded, we can accept the current values of $\beta$ and proceed to the inference of angular coordinates. Otherwise, $\beta$ needs to be corrected and the hidden degrees must be recomputed. Since the expected mean local clustering coefficient is a monotonic function of $\beta$, the process can be efficiently evaluated using the bisection method. More specifically, we start with a value of $\beta$ chosen randomly between $D$ and $D + 1$. Then, while the expected clustering is lower than the observed one, we multiply $\beta$ by 1.5. We start the bisection method when we reach the value for which the observed clustering is surpassed. We found that for $\epsilon_{\bar{c}} = 0.01$, $m = 600$ is enough. To gain higher precision, one must increase $m$ to ensure that the required precision is satisfied.

**$\mathbb{S}^D$ model-corrected Laplacian Eigenmaps.** The expected angular distance between nodes $i$ and $j$ in the $\mathbb{S}^D$ model, conditioned to the fact that they are connected, can be computed as

$$\langle\Delta\theta_{ij}\rangle = \int_0^\pi \Delta\theta_{ij}\rho(\Delta\theta_{ij}|a_{ij} = 1)\,d\Delta\theta_{ij} \tag{19}$$

$$= \frac{\int_0^\pi \frac{\Delta\theta\sin^{D-1}\Delta\theta \, d\Delta\theta}{1 + \left(\frac{R\Delta\theta}{(\mu\kappa(k_i)\kappa(k_j))^{1/D}}\right)^\beta}}{\int_0^\pi \frac{\sin^{D-1}\Delta\theta d\Delta\theta}{1 + \left(\frac{R\Delta\theta}{(\mu\kappa(k_i)\kappa(k_j))^{1/D}}\right)^\beta}}. \tag{20}$$

Since degree-one nodes do not add geometric information, we first obtain the positions for nodes with $k > 1$, and subsequently

reincorporate the nodes with degree of one. For each node $i$ with $k_i = 1$ and its neighbor $j$, we draw an angular distance $\Delta\theta_{ij}$ from Eq. (14) given that the two connected nodes have hidden degrees $\kappa_i$ and $\kappa_j$. Then, the position of node $i$, $\mathbf{v}_i$, is generated with a given angular separation to the node $j$.

**Likelihood maximization.** Given initial positions for the nodes on the $D$-sphere, the steps to maximize the congruency between the observed network and the $\mathbb{S}^D$ model are:

1. *Define an ordering of nodes*: The nodes are visited in the order defined by the networks' onion decomposition[72]. In the sequence, the ordering of nodes belonging to the same layer in the decomposition is random.
2. *Find new optimal coordinates*: For every node $i$, we select the optimal coordinates among candidates' positions generated in the vicinity of the mean vector of its neighbors. This is achieved in three steps:
   (a) *Compute the mean coordinates of node i's neighbors*. Let node $i$ have $k_i$ neighbors, which are now labeled with index $j = 1, ..., k_i$. Since the nodes are situated on the $D$-sphere we have to compute their mean vector $\bar{\mathbf{v}}_i$, which is given by

$$\bar{\mathbf{v}}_i = \sum_j \frac{1}{\kappa_j^2} \mathbf{v}_j \qquad (21)$$

where the hidden degrees in the above expression weight the contribution of every neighbor's positioning vector, as proposed in ref. 25.
   (b) *Propose new positions around $\bar{\mathbf{v}}_i$*: We generate $100\max(\ln N, 1)$ candidate vectors from the multivariate normal distribution with mean $\bar{\mathbf{v}}_i$ and standard deviation $\sigma$ given by

$$\sigma = \max\left(\frac{\pi}{2}, \frac{\Delta\theta_{\max}}{2}\right), \qquad (22)$$

where $\Delta\theta_{\max}$ is the angular distance between vector $\bar{\mathbf{v}}_i$ and the most distant neighbor of node $i$.
   (c) *Select the most likely candidate position*: Compute the local log-likelihood of every candidate position as well as of node $i$'s current position according to

$$\ln\mathcal{L}_i = \sum_{i\neq j} a_{ij}\ln p_{ij} + (1 - a_{ij})\ln(1 - p_{ij}) \qquad (23)$$

Locate node $i$ at the position maximizing the local log-likelihood.

**Final adjustment of hidden degrees.** The process of adjusting hidden degrees, given the positions of the nodes in the similarity subspace, such that $\bar{k}(\kappa_i) = k_i$ is similar to the initial inference of hidden degrees:

1. *Compute the expected degrees*: For every node $i$, set

$$\bar{k}(\kappa_i) = \sum_{i\neq j} \frac{1}{1 + \left(\frac{R\Delta\theta_{ij}}{(\mu\kappa_i\kappa_j)^{1/D}}\right)^\beta}. \qquad (24)$$

2. *Correct hidden degrees*: Let $\varepsilon_{\max} = \max_i\{|\bar{k}(\kappa_i) - k_i\}$ be the maximal deviation between degrees and expected degrees. If $\varepsilon_{\max} > \varepsilon$, the set of the hidden degrees needs to be corrected. Then set $|\kappa_i - [k_i - \bar{k}(\kappa_i)]u| \to \kappa_i$ for every node $i$, where $u \sim U(0, 1)$. Again, the random variable $u$ prevents the process from getting trapped in the local minimum. Next, go to step 1 and compute the expected degrees corresponding to the new set of hidden degrees. Otherwise, if $\varepsilon_{\max} < \varepsilon$, the hidden degrees have been inferred for the current global parameters and nodes' positions.

## Generating synthetic networks with the $\mathbb{S}^D$ model

1. The distribution of hidden degrees can be of any form. For the experiments in Figs. 2 and 3, we used a power-law hidden degree distribution of the form $\rho(\kappa) = (\gamma - 1)\kappa_0^{\gamma-1}\kappa^{-\gamma}$, with $\kappa > \kappa_0 = (\gamma - 2)/(\gamma - 1)\langle k\rangle$ and different values of the characteristic exponent $2 < \gamma < 3$. Hidden degrees are also cut off from above by the natural cut-off $\kappa_c = \kappa_0 N^{\frac{1}{\gamma-1}}$[73]. This choice avoids the extreme fluctuations of the maximum hidden degrees for $\gamma < 3$ that would result, for some network realizations, in expected degrees larger than $N$. For every node $i$ in the simulated network, we generated the hidden degree $\kappa_i$ as a random value from this distribution.
2. To assign nodes' positions on the similarity subspace, each node is assigned a vector $\mathbf{v}_i \in \mathbb{R}^{D+1}$ with $D + 1$ independent and standard normally distributed entries. These entries are subsequently normalized to lie on the sphere with $||\mathbf{v}_i|| = R$, where $R$ is given in Eq. (2).
3. The hidden degrees and the coordinates in the $D$-dimensional similarity subspace are used in Eq. (1) to calculate the probability of connection between any pair of nodes. For any given value of $\beta$, the value of $\mu$ is evaluated from Eq. (3) depending on the target average degree $\langle k\rangle$, $\beta$, and $D$.

### Reporting summary

Further information on research design is available in the Nature Portfolio Reporting Summary linked to this article.

### Data availability

The network datasets used in this study are available from the sources referenced in the manuscript and the supplementary materials. The coordinates and the parameters of the multidimensional hyperbolic embeddings of the real networks are available via the Zenodo platform at https://doi.org/10.5281/zenodo.10027084.

### Code availability

The code of the $D$-Mercator embedding tool is publicly available at https://github.com/networkgeometry/d-mercator.

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

## Acknowledgements

M.A.S. and M.B. acknowledge support from grant TED2021-129791B-I00 funded by MCIN/AEI/10.13039/501100011033 and the "European Union NextGenerationEU/PRTR", grant PID2019-106290GB-C22 funded by MCIN/AEI/10.13039/501100011033, and the Generalitat de Catalunya grant number 2021SGR00856. R.J. acknowledges support from the fellowship FI-SDUR funded by Generalitat de Catalunya. M.B. acknowledges the ICREA Academia award, funded by the Generalitat de Catalunya. A.A. acknowledges financial support from the Sentinelle Nord initiative of the Canada First Research Excellence Fund and from the Natural Sciences and Engineering Research Council of Canada (project 2019-05183).

## Author contributions

M.A.S. designed research; R.J., M.B., and M.A.S. implemented research; R.J. and A.A. generated the code. R.J. performed the computations and numerical simulations. All authors discussed the results and implications and contributed to the writing of the manuscript.

## Competing interests

The authors declare no competing interests.
