## [Peer Review File · Nature Communications]

REVIEWER COMMENTS

Reviewer #1 (Remarks to the Author):

In this paper the authors describe a connection-model-based method for embedding nodes of a network into a D -sphere similarity space, with additional dimension for effectively correcting by degree. They describe the method, validate that the method works for synthetic networks, demonstrate relationships to navigability and communities, and then apply the method to a small collection of "real world networks". Importantly, the authors also provide code to implement their method (though I have not attempted to use their code so I am merely trusting that it is indeed reasonable to use).

While the overall direction of the results here seem reasonable enough, I am having a very difficult time coming up with a justification for why this paper should be published in Nature Communications. Among other issues, the current version of the manuscript is not in the regular form of articles in the journal, which makes assessing the quality of the possibly final exposition very difficult. Specifically, when the authors end up rewriting everything into the standard headings of 'Results' and 'Methods', will the motivation and purpose of this paper be clearer? The current version of the manuscript was very clearly originally written for a different journal. It has an only very terse introduction that already presumes a lot of background knowledge by the reader and presumes that the reader is already motivated to want to perform network embeddings, with only very quickly strung together citations of networks topics and papers where embedding was important. It then dives into the methods (though not labeled as such) right there on the first page. The most compelling to me language about why we might want to look at embeddings doesn't appear until the Conclusion.

Meanwhile, the only 'Results' per se appear to be those demonstrated in Figures 3, 4 and 5, and only 1 of those figures (Fig.5) is on data not generated by the same underlying model. That is, Figure 2 is only a validation that the inference procedure works when applied to data generated from the model used here. Figure 3 demonstrates the success of greedy routing in the multidimensional space for synthetic data works best when the embedding space used for fitting is the same as that used to generate the synthetic data. Figure 4 similarly compares the $D=1$ and $D=2$ embeddings for community concentration on synthetic data from these two dimensions. Then Figure 5 repeats the greedy routing and community concentration calculations for different embedding dimensions for 6 "real world networks". Even then, what do you really learn from these embeddings here? Knowing the dimension of the best embedding is presented as somehow essential information with only very limited discussion about what knowing it does for the scientist analyzing the data. What did you really learn about these 6 networks here that wasn't already apparent from the definitions or some other method? The most dramatic statement about a result in Section VI might be "the topology of the airports network is 1-dimensional in the Euclidean similarity space whereas one could have naively expected $D = 2$. This last result highlights the distinction between the geometry of the Earth and that of the topology of the airports network, where

long-range flights between hubs reduce the effective geometry of the planet." What else did we really learn from this analysis?

The method itself is perhaps technically interesting to a specific subset of experts in the network science community. The journal's Aims & Scope declares that "Papers published by the journal aim to represent important advances of significance to specialists within each field." Does this really rise to "important" and "of significance"? To me it is definitely technically interesting, but I'd need to see much more, especially much more learned about many more real world networks, to convince me that this method is truly useful in general for analyzing real world networks before I can recommend its publication in Nature Communications. Instead, I recommend that the authors reconsider submitting their work to a more networks-specialized journal.

Reviewer #2 (Remarks to the Author):

This well-written paper introduces a novel algorithm, namely D-Mercator, for embedding networks in $D+1$ hyperbolic spheres. The algorithm is a generalization of the popular method Mercator, previously developed for $D=1$ only. The new algorithm is validated with extensive experiments in synthetic graphs. The application of the algorithm to real networks clearly motivates the need to embed networks in $D>1$ (thus the necessity of deploying a novel algorithm for this task). I liked the paper, and I am favor of its publication in the journal. I have only minor comments that the authors can optionally consider in a revised version of their manuscript:

1. In the caption of Table I, the authors write "... D is the inferred dimension as determined by the performance of greedy routing and community concentration." This is bit vague, as it leaves room for multiple interpretations. Looking at the SI, it seems to me that the authors did perform a qualitative determination of the "optimal" D value. For example, in Polbooks $D = 2$ is preferred to $D = 1$ only because of performance in greedy routing, whereas $D=1$ and $D=2$ are similar in performance regarding community concentration; for Foxglove, the opposite is true, as $D=2$ and $D=3$ yield similar performance in greedy routing but $D=3$ seems better than $D=2$ in determining community concentration. I believe the authors should be a little more precise in describing the details of this part of their analysis.

2. Still in relation with point 1, I am a little hesitant in thinking of the performance of community concentration as a good criterion to decide the value of the optimal embedding dimension of a real

network. The main issue is that the "ground-truth" communities that the authors define on the basis of some meta data are not necessarily reflected as good communities in the structure of the network. Empirical evidence of this statement is offered in some existing publications, e.g., Phys. Rev. E 90, 062805 (2014). Performance in greedy routing is certainly a more objective criterion. However, other equally objective criteria, e.g., performance in link prediction, can be formulated. These may lead to different estimates of the optimal D value. I believe that a discussion on this issue would be beneficial for the interpretation of the results on real-world networks.

Reviewer #3 (Remarks to the Author):

This is a review for manuscript 'D-Mercator: multidimensional hyperbolic embedding of real networks' by R. Jankowski et al., which has been submitted for consideration in Nature Communications with ID NCOMMS-23-15745.

In this paper the authors introduce D-Mercator, an extension of previous network embedding techniques that maps networks into the multidimensional, $D+1$ hyperbolic space, formed by 1- and D -dimensional (spherical) subspaces representing hidden degrees (a measure of node importance) and node similarities, respectively. The algorithm D-Mercator takes as input a network and uses two techniques, model-based Laplacian eigenmaps and maximum likelihood estimation, to infer node coordinates in hyperbolic space, their hidden degrees, plus a model parameter β regulating clustering in the network.

These maps are arguably more informative than two-dimensional embeddings found in the literature, particularly in identifying factors that determine network connectivity, and in estimating the optimal dimensionality of network embeddings with respect to navigability and community structure.

The paper is interesting, very well written, and describes a nice technical addition to the dynamic field of network geometry. Perhaps my biggest worry is whether it contains sufficient phenomenological breakthroughs as to appeal to the broader audience of Nat. Comms. Research on the embedding of complex networks into hyperbolic spaces (and the consequences of that mapping into our understanding of networks) is a rich and burgeoning field, as indicated by many of the publications of the authors themselves in the last 15 years. This paper follows such line by extending a previous network embedding method to multiple dimensions, and showing how the optimal dimension of that embedding (typically a small number for the empirical datasets considered) correlates with optima of navigability and modular structure detection in the original network. Apart from the clear usefulness of low-dimensionality network visualisations, the paper stops there and doesn't delve further on the benefits of this technical development to our understanding of networks, or on the consequences of the

apparent low dimensionality of hyperbolic embeddings of empirical networks. Particularly, comparing with the previous and recent work in Ref. [22] by the authors, it feels like the phenomenological breakthrough that real networks can be associated to a low dimension was explored there, and this paper represents a technical advance on how to confirm those results via spatial embedding. In that sense, I think this paper might be suitable for a more specialised journal, and for an audience well-versed on the details and methods of network geometry.

In any case, here are a few comments on the technical aspects of the work, which I hope might help the authors increase the quality of their manuscript even further.

TECHNICAL COMMENTS

- Sec. III (after Eq. 3) and Sec. IV (paragraph 3): If the hidden degrees are estimated iteratively from an initial guess equal to the observed degrees in the original network, why do you write in Sec. III (after Eq. 3) that ‘... The hidden degrees can be generated randomly from an arbitrary distribution or taken as a set of prescribed values’? At this point in the text this phrase might be a bit confusing.

- Sec. IV (paragraph 3): Initial guess for β : How is the initial guess for the inverse temperature β selected? Is the initial guess chosen depending on some property of the original network, such as clustering coefficient? Does the quality of the inference method depend on the value of this initial guess? After I while I found details on this at the end of Appendix B2 (a random value between D and $D+1$, improved via bisection with expressions for the clustering coefficient). Perhaps this could be stated, briefly but explicitly, in Sec. IV (paragraph 3) for improved clarity in the method.

- Sec IV and Appendix B4: The fine-tuning of inferred node coordinates by MLE described in Appendix B4 (order nodes by onion decomposition, calculate mean position of neighbours, generate Gaussian-distributed candidate positions, select the one that maximises local log-likelihood), seems a bit idiosyncratic as a heuristic and composed of several moving parts. How did the authors arrive at this procedure? (I guess algorithm scalability might be part of the answer...). Do these inferred node coordinates correspond to the global optimum maximising the likelihood that the hyperbolic model reproduces the original network? Or can it be trapped in some local optima? The authors do indeed validate D-Mercator via synthetic networks (Sec. V), but I wonder if they could say a few words about alternative, more global way of inferring model parameters.

- Sec. V A and Fig. 2: Validation via synthetic networks: The validation process (consistency of estimators) described in Sec. V requires generating networks with the S^D model (i.e. specifying hidden

degrees/coordinates and β) and then fitting them back to the model (potentially with a different dimension) via the embedding method, ideally recovering the same values. In Fig. 2, the authors write that S^2 synthetic networks follow a power-law degree distribution with exponent γ . How is γ related to the definition of the hyperbolic model in, e.g., Eqs. 1-3? Are the hidden degrees κ power-law distributed according to γ , and the node coordinates distributed uniformly at random over the D-dimensional sphere? The description of synthetic network generation in Sec. V could be made clearer.

- Sec. V C and Fig. 4: Synthetic networks with modular structure: How much community overlap is there as a function of the polar angle parameter? Does overlap affect the ability of the embedding method to recover the geometric concentration of communities?

MINOR COMMENTS

- Introduction, paragraph 2: ... directed network(s)... is there a missing 's'?

- After Eq. 8: Is $n_{i,j}$ actually $n_{i,l}$?

Reviewer #1

Comment 1.0.:

In this paper the authors describe a connection-model-based method for embedding nodes of a network into a D-sphere similarity space, with additional dimension for effectively correcting by degree. They describe the method, validate that the method works for synthetic networks, demonstrate relationships to navigability and communities, and then apply the method to a small collection of "real world networks". Importantly, the authors also provide code to implement their method (though I have not attempted to use their code so I am merely trusting that it is indeed reasonable to use).

Reply: We thank the referee for their time, for their thorough evaluation of our paper, and for providing constructive feedback that helped us improve the quality of our work. After this resubmission, we hope the referee will recognize the value of our work and deem it suitable for publication in Nature Communications. Moving forward, we will address individually each comment raised by the referee.

Comment 1.1.:

While the overall direction of the results here seem reasonable enough, I am having a very difficult time coming up with a justification for why this paper should be published in Nature Communications. Among other issues, the current version of the manuscript is not in the regular form of articles in the journal, which makes assessing the quality of the possibly final exposition very difficult. Specifically, when the authors end up rewriting everything into the standard headings of 'Results' and 'Methods', will the motivation and purpose of this paper be clearer? The current version of the manuscript was very clearly originally written for a different journal. It has an only very terse introduction that already presumes a lot of background knowledge by the reader and presumes that the reader is already motivated to want to perform network embeddings, with only very quickly strung together citations of networks topics and papers where embedding was important. It then dives into the methods (though not labeled as such) right there on the first page. The most compelling to me language about why we might want to look at embeddings doesn't appear until the Conclusion.

Reply: We partly agree with the referee's comments and acknowledge that we did not clearly explain the relevance of our work in the introduction. Over the past decade, we have been actively researching the emerging field of network geometry, and during this time, we have discovered numerous empirical pieces of evidence supporting the idea that real networks exhibit "geometric" characteristics. By this, we do not simply mean that these networks can be embedded in a metric space, but rather that distances within this latent metric space play a crucial role in establishing connections within the network.

This observation led us to conclude that by learning the coordinates in this space, we can construct a map of the network. Such maps have proven to be valuable in various applications, including information routing, node classification and link prediction, hierarchy definition, and renormalization, among others. However, it is important to note that maps are only useful if they accurately represent the original network structure. When we project a higher-dimensional object onto a lower-dimensional

space, we inevitably lose information, as components of the system that were originally separate can appear together in the lower-dimensional representation.

In a recent publication, we made the intriguing discovery that real networks predominantly reside in ultra low-dimensional hyperbolic spaces, but often exceeding the two dimensions commonly utilized in previous studies. The significance of our work lies not only in determining the most plausible dimension but also in finding the embedding of the system within this higher-dimensional space. Through this process, we enhance our resolution power, enabling us to differentiate previously blended components that now appear distinct. This becomes very clear in the case of the Add-health network, which we include in the main paper as a case example. For this network, embedding it in the two-dimensional hyperbolic plane, and thus having a one-dimensional similarity subspace, leads to the mixing of several ground-truth communities. For instance, the 7th and 8th grades, which should appear well-separated as expected, instead become mixed. However, if the embedding is performed in a two-dimensional similarity space, they appear well separated, as they should be. This example illustrates very clearly that by choosing the wrong dimension, one can end up with an inaccurate representation of the system.

Of course, please note that this is one example among many, but it highlights the importance of selecting the appropriate dimension to obtain a meaningful representation of the data. To summarize, our paper presents the development and implementation of an algorithm that generates highly informative maps of real complex networks.

Action taken: We have revised and improved the introduction and restructured the organization of the paper to better convey the points mentioned above. For the sake of clarity, in the new version of the manuscript, we focus only on one case example, the Add-health network, and moved the rest of real networks to the supplementary information.

Comment 1.2.:

Meanwhile, the only 'Results' per se appear to be those demonstrated in Figures 3, 4 and 5, and only 1 of those figures (Fig.5) is on data not generated by the same underlying model. That is, Figure 2 is only a validation that the inference procedure works when applied to data generated from the model used here. Figure 3 demonstrates the success of greedy routing in the multidimensional space for synthetic data works best when the embedding space used for fitting is the same as that used to generate the synthetic data. Figure 4 similarly compares the $D=1$ and $D=2$ embeddings for community concentration on synthetic data from these two dimensions. Then Figure 5 repeats the greedy routing and community concentration calculations for different embedding dimensions for 6 "real world networks". Even then, what do you really learn from these embeddings here? Knowing the dimension of the best embedding is presented as somehow essential information with only very limited discussion about what knowing it does for the scientist analyzing the data. What did you really learn about these 6 networks here that wasn't already apparent from the definitions or some other method? The most dramatic statement about a result in Section VI might be "the topology of the airports network is 1-dimensional in the Euclidean similarity space whereas one could have naively expected $D = 2$. This last result highlights the distinction between the geometry of the Earth and that of the topology of the airports network, where long-range flights between hubs reduce the effective geometry of the planet." What else did we really learn from this analysis?

Reply: As the referee correctly pointed out, the purpose of figures 2 to 4 was to assess the accuracy of the embedding algorithm presented in this work and to examine methods for detecting the optimal dimension based on the embeddings. We consider this step to be crucial since, without successful results on synthetic networks generated from the model, the application of the method to real networks would not be reliable. With that in mind, in figure 5, we applied the same methodology to real networks,

confident that our model accurately captures the underlying topologies of these networks. The real networks depicted in this figure were selected because they are highly suited for representation in three-dimensional hyperbolic space. Consequently, the similarity space takes the form of a two-dimensional sphere, enabling us to visualize the network embedding. Nevertheless, as previously stated, research on network geometry has demonstrated that our geometric models can effectively describe a wide range of real networks. Hence, the embedding method introduced in this paper holds potential for application to numerous diverse real systems.

Regarding the comment on the insights gained from our analysis of these networks, we would like to emphasize that it is not solely determining the dimension that holds significance, but rather finding the appropriate embedding in that dimension. This crucial aspect allows us to generate an informative and precise map of each system. Once we have obtained the optimal map, it can be employed for various applications, akin to the utilization of maps in other contexts where they serve to convey pertinent information. In this resubmission, we have focused on the Add-health network as a case example. For this network, it is evident that using a one-dimensional embedding in a similarity space leads to inaccurate results. This is particularly noticeable in the 7th and 8th grades, where they appear mixed up in the one-dimensional embedding, whereas they are clearly separated in the two-dimensional embedding. The latter result aligns with expectations since junior students in the early high school grades did not have the opportunity to interact with higher grades.

Action taken: We have reorganized the figures and encapsulated figures 3 to 4 into a single panel showing the results on synthetic networks. We have added the results of applying community detection to real networks and have separated figure 5 into three figures for each dataset. The results for Add-health are shown in the main text as a case study. We have extended the discussion of the obtained real network embeddings and we have clarified the importance of identifying correctly the best embedding dimension for every real network.

Comment 1.3.:

The method itself is perhaps technically interesting to a specific subset of experts in the network science community. The journal's Aims & Scope declares that "Papers published by the journal aim to represent important advances of significance to specialists within each field." Does this really rise to "important" and "of significance"? To me it is definitely technically interesting, but I'd need to see much more, especially much more learned about many more real world networks, to convince me that this method is truly useful in general for analyzing real world networks before I can recommend its publication in Nature Communications. Instead, I recommend that the authors reconsider submitting their work to a more networks-specialized journal.

Reply: Beyond the technical interest of the paper, that as the referee recognizes is definitely significant for specialists, identifying the correct dimensionality of a real network is crucial in many contexts and recurs across disciplines in the natural sciences, from the string theory framework for particle physics and quantum gravity to statistical physics. For instance, in statistical physics the number of spatial dimensions is one of the few factors that determines the critical properties and the universality class of extended systems where events at multiple length scales make relevant contributions. Even if in network science we do not have yet a comprehensive theory for critical phenomena, the dimensionality of network structure will definitely be one of the relevant variables in such a theory. We believe that the possibility of a multidimensional analysis of real networks within the network geometry framework represents a huge step forward into that direction. In addition, the two-dimensional maps produced by the original version of the embedding tool using a one-dimensional similarity space have already proved to be extremely good in describing connectivity patterns in real networks. Using the right dimensionality brings an improvement in the accuracy and resolution that is badly needed for many

downstream tasks. We hope that the new version of the manuscript is more clear and convincing.

Action taken: We restructured the organization of the paper, revised the introduction, emphasized the significance of our work, added more analysis on the results about the real networks reported in the previous version of the manuscript, and incorporated a community detection task, see answers to points 1.1 and 1.2..

Reviewer #2

Comment 2.0:

This well-written paper introduces a novel algorithm, namely D-Mercator, for embedding networks in $D+1$ hyperbolic spheres. The algorithm is a generalization of the popular method Mercator, previously developed for $D=1$ only. The new algorithm is validated with extensive experiments in synthetic graphs. The application of the algorithm to real networks clearly motivates the need to embed networks in $D>1$ (thus the necessity of deploying a novel algorithm for this task). I liked the paper, and I am favor of its publication in the journal.

Reply: We thank the reviewer for their time and effort in reading our paper and for their report, which has helped us to improve substantially the presentation of our results. We are delighted to learn that the reviewer considers our paper well-written and deems it suitable for publication in Nature Communications.

Comment 2.1:

I have only minor comments that the authors can optionally consider in a revised version of their manuscript:

1. In the caption of Table I, the authors write "... D is the inferred dimension as determined by the performance of greedy routing and community concentration." This is bit vague, as it leaves room for multiple interpretations. Looking at the SI, it seems to me that the authors did perform a qualitative determination of the "optimal" D value. For example, in Polbooks $D = 2$ is preferred to $D = 1$ only because of performance in greedy routing, whereas $D=1$ and $D=2$ are similar in performance regarding community concentration; for Foxglove, the opposite is true, as $D=2$ and $D=3$ yield similar performance in greedy routing but $D = 3$ seems better than $D=2$ in determining community concentration. I believe the authors should be a little more precise in describing the details of this part of their analysis.

Reply: We agree with the referee that an explicit statement about the prescription used to determine the inferred dimension of a network from the performance of greedy routing and community concentration was missing in the manuscript. Nevertheless, we value versatility interpreted as the quality of being useful for different purposes and from this point of view the results are quite obvious. For example, in Polbooks $D = 2$ is preferred to $D = 1$ not only because of performance in greedy routing, the performance in community concentration supports the selection since the best dimensions in community concentration are $D=1$ and $D=2$ and any of the two produce the best outcome, but not higher dimensions. Thus, in this case greedy routing disambiguates community concentration while being perfectly consistent. The same for Foxglove, as $D=2$ and $D=3$ yield similar performance in greedy routing, community concentration is used to disambiguate giving $D=3$. There is always an

agreement, and never a contradiction, in the results for greedy routing and community concentration. To strengthen the results, we have implemented an agglomerative clustering algorithm to find a community partition of the network based on the distances in the similarity space from embeddings in different dimensions. We then measured the normalized mutual information between such partitions and the ground-truth metadata and also the modularity of the partition. As it can be seen in the new Fig 4, both measures corroborates our previous findings.

Action taken: In the main text of the manuscript, we have clarified the criterium to determine the inferred dimension. We have also added the results of community detection using agglomerative clustering algorithms, supporting our previous findings.

Comment 2.2:

2. Still in relation with point 1, I am a little hesitant in thinking of the performance of community concentration as a good criterion to decide the value of the optimal embedding dimension of a real network. The main issue is that the "ground-truth" communities that the authors define on the basis of some meta data are not necessarily reflected as good communities in the structure of the network. Empirical evidence of this statement is offered in some existing publications, e.g., Phys. Rev. E 90, 062805 (2014). Performance in greedy routing is certainly a more objective criterion. However, other equally objective criteria, e.g., performance in link prediction, can be formulated. These may lead to different estimates of the optimal D value. I believe that a discussion on this issue would be beneficial for the interpretation of the results on real-world networks.

Reply: The referee is right and we totally agree in this point. Communities on the basis of some metadata are not necessarily reflected as good communities in the structure of a network. Nevertheless, it is still remarkable that, as explained in the previous answer, there is always an agreement, and in any case never a contradiction, in the results for greedy routing and community concentration. We have also evaluated the community detection task where the labels of nodes were inferred from the similarity space, corroborating our previous findings.

Action taken: We have also added the results of a new community detection task, that support our previous findings.

Reviewer #3

Comment 3.0:

This is a review for manuscript 'D-Mercator: multidimensional hyperbolic embedding of real networks' by R. Jankowski et al., which has been submitted for consideration in Nature Communications with ID NCOMMS-23-15745.

In this paper the authors introduce D-Mercator, an extension of previous network embedding techniques that maps networks into the multidimensional, $D+1$ hyperbolic space, formed by 1- and D -dimensional (spherical) subspaces representing hidden degrees (a measure of node importance) and node similarities, respectively. The algorithm D-Mercator takes as input a network and uses two techniques, model-based Laplacian eigenmaps and maximum likelihood estimation, to infer node coordinates in hyperbolic space, their hidden degrees, plus a model parameter β regulating clustering in the network.

These maps are arguably more informative than two-dimensional embeddings found in the literature, particularly in identifying factors that determine network connectivity, and in estimating the optimal dimensionality of network embeddings with respect to navigability and community structure.

The paper is interesting, very well written, and describes a nice technical addition to the dynamic field of network geometry.

Reply: We thank the reviewer for their time and effort in evaluating our paper, and for their report. We are glad to learn that the reviewer considers our paper interesting and very well-written. The new version of the manuscript has been improved substantially following their comments and the comments by all the referees. We hope that now our work is more clear about the significance of having network maps in the right dimension.

Comment 3.1:

Perhaps my biggest worry is whether it contains sufficient phenomenological breakthroughs as to appeal to the broader audience of Nat. Comms. Research on the embedding of complex networks into hyperbolic spaces (and the consequences of that mapping into our understanding of networks) is a rich and burgeoning field, as indicated by many of the publications of the authors themselves in the last 15 years. This paper follows such line by extending a previous network embedding method to multiple dimensions, and showing how the optimal dimension of that embedding (typically a small number for the empirical datasets considered) correlates with optima of navigability and modular structure detection in the original network. Apart from the clear usefulness of low-dimensionality network visualisations, the paper stops there and doesn't delve further on the benefits of this technical development to our understanding of networks, or on the consequences of the apparent low dimensionality of hyperbolic embeddings of empirical networks. Particularly, comparing with the previous and recent work in Ref. [22] by the authors, it feels like the phenomenological breakthrough that real networks can be associated to a low dimension was explored there, and this paper represents a technical advance on how to confirm those results via spatial embedding. In that sense, I think this paper might be suitable for a more specialised journal, and for an audience well-versed on the details and methods of network geometry.

Reply: Notice that the phenomenological breakthrough in Ref. [22] was a crucial step in the portray of the multidimensional description of real networks, but the question goes a lot beyond. In Ref. [22], we proved that real networks can be associated to ultra low dimensions in the hyperbolic description, but nothing was said about the coordinates of the nodes in such multidimensional spaces. We were not providing the methodology to reconstruct the multidimensional maps of real networks, and without such knowledge, downstream theoretical and practical applications were impossible. In the current work, we fill the gap and introduce the methodology to produce multidimensional embeddings. So we go from knowing just the appropriate dimension to know not only the dimension but all the distances and relative positions between all pairs of nodes.

The usefulness of low-dimensional network visualizations is clear, but our results are not only visually appealing but the obtained coordinates and distances are also meaningful. This is what allowed us to delve further on the benefits of this technical development. As the referee points out, in the previous version of the manuscript, we showed how the optimal dimension of the embedding (typically a ultra small number for the empirical datasets considered) correlated with optima of navigability and with modular structure defined by metadata in the original network. This not only confirms the results in [22] via spatial embedding, but the multidimensional spatial embedding helps us to discern new information. In our manuscript, we already showed that an embedding in the right dimension is fundamental to have enough resolution to discern ground-truth communities, as shown in Fig. 3, and that groups defined by metadata in real networks localize in similarity space, as shown in Fig. 4.

This findings suggest that community detection, a broad and important topic in network science and related fields, can be one of the great beneficiaries of multidimensional maps of real networks. In the current version of the manuscript, we have added community detection experiments on real networks based on their multidimensional geometric layout. Specifically, we have implemented an agglomerative clustering algorithm to find a partition of the network based on the distances in the similarity space from embeddings in different dimensions. We then measure the normalized mutual information between such partitions and the ground-truth metadata and also the modularity of the partition. As it can be seen in the new Fig 4, both measures corroborates our previous findings.

Action taken: We have extended the discussion of the obtained real network embeddings. We have also added the results of detecting communities in real networks based on their multidimensional embeddings, that support our previous findings and help clarifying the benefits of the multidimensional embedding technique.

Comment 3.2: In any case, here are a few comments on the technical aspects of the work, which I hope might help the authors increase the quality of their manuscript even further.

TECHNICAL COMMENTS

- Sec. III (after Eq. 3) and Sec. IV (paragraph 3): If the hidden degrees are estimated iteratively from an initial guess equal to the observed degrees in the original network, why do you write in Sec. III (after Eq. 3) that '... The hidden degrees can be generated randomly from an arbitrary distribution or taken as a set of prescribed values'? At this point in the text this phrase might be a bit confusing.

Reply: Note that the $S/1H^{i+1}$ model can be used as topology generator (we like to say going from geometry to topology), or to infer network maps (the other way around, going from topology to geometry). In Sec. III, we summarize the $S/1H^{i+1}$ model, while in Sec. IV we explain the multidimensional network embedding method based on the model described in Sec. III. The model can be used independently of any embedding technique and, as a generative model, independently of real networks. It is general and, as said in the text, the hidden degrees can be generated randomly from an arbitrary distribution or taken as a set of prescribed values, akin for instance to the configuration model (in fact, the $S/1H^{i+1}$ model converges to the configuration model in the limit $\beta \rightarrow 0$). In contrast, the inference problem solved by D-Mercator in Sec. IV aims at finding the coordinates of the nodes in a real network that maximize the likelihood that the observed network topology is generated by the model. In this case, a clever initial guess for the hidden degrees is given by the observed degrees in the original network. The reason being that, in the model, the expected degree of a node with hidden variable i is, precisely, $k^\downarrow(i) = i$, as explained at the end of Sec. III.

Comment 3.3:

- Sec. IV (paragraph 3): Initial guess for β : How is the initial guess for the inverse temperature β selected? Is the initial guess chosen depending on some property of the original network, such as clustering coefficient? Does the quality of the inference method depend on the value of this initial guess? After I while I found details on this at the end of Appendix B2 (a random value between D and $D+1$, improved via bisection with expressions for the clustering coefficient). Perhaps this could be stated, briefly but explicitly, in Sec. IV (paragraph 3) for improved clarity in the method.

Reply: We agree with the referee that choice of the initial value of the inverse temperature β was not clearly explained in the main text. In the model, $\beta = D$ marks a phase transition between the geometric and the non-geometric regimes (see Ref.[9] in the manuscript) and $\beta = D + 1$ separates the small-world regime from a regime in which the network is small-world only if the degree distribution is

heterogeneous with $\gamma < 3$. Hence, the initial value of β was chosen randomly between D and $D + 1$ to ensure that the model is in the geometric regime $\beta > D$. Note, however, that the quality of the inference method does not depend on the value of this initial guess as the final inferred value can take any value between D and infinity. Notice also that the inference of parameter β is coupled to the inference of the nodes' hidden degrees \mathbf{h} , so that the final combination of parameters β and \mathbf{h} is such that the degree distribution and clustering coefficient are both correctly reproduced by the geometric model.

Action taken: We added a sentence to the main text to clarify the choice of the initial value of β .

Comment 3.4:

- Sec IV and Appendix B4: The fine-tuning of inferred node coordinates by MLE described in Appendix B4 (order nodes by onion decomposition, calculate mean position of neighbors, generate Gaussian-distributed candidate positions, select the one that maximizes local log-likelihood), seems a bit idiosyncratic as a heuristic and composed of several moving parts. How did the authors arrive at this procedure? (I guess algorithm scalability might be part of the answer). Do these inferred node coordinates correspond to the global optimum maximizing the likelihood that the hyperbolic model reproduces the original network? Or can it be trapped in some local optima? The authors do indeed validate D-Mercator via synthetic networks (Sec. V), but I wonder if they could say a few words about alternative, more global way of inferring model parameters.

Reply: As the referee correctly points out, and as was stated in Sec. II and Sec. IV of the previous version of the manuscript, the Mercator family of embedding models try to find the global optimum maximizing the likelihood that the hyperbolic model reproduces the original network. However, the problem, as the majority of optimization and network inference problems, is NP hard and the solution can be trapped in some local optima. This is what makes particularly important a proper validation protocol on the basis of synthetic networks produced by the model such that the ground truth is known and the results can be compared against it.

Even if alternative embedding methods have been proposed to map hyperbolic networks in $D = 1$, Mercator is an accurate mapping tool that outperforms other embedding algorithms. Apart from its accuracy, Mercator has the advantage of systematically inferring not only nodes coordinates but also global model parameters, and has the ability to embed networks with arbitrary degree distributions in reasonable computational time, which makes it competitive for real applications. D-Mercator is our proposed solution based on previous experience of network embedding in $D = 1$. Nevertheless, we believe that machine learning techniques will provide in the future a complementary framework for inferring nodes' coordinates and global model parameters defining the global optimum that maximizes the likelihood that the hyperbolic model reproduces the original network.

Action taken: We have added a few words to the Conclusion section about future avenues for alternative techniques in model-based hyperbolic network embeddings.

Comment 3.5:

- Sec. V A and Fig. 2: Validation via synthetic networks: The validation process (consistency of estimators) described in Sec. V requires generating networks with the S^D model (i.e. specifying hidden degrees/coordinates and β) and then fitting them back to the model (potentially with a different dimension) via the embedding method, ideally recovering the same values. In Fig. 2, the authors write that S^2 synthetic networks follow a power-law degree distribution with exponent γ . How is γ related to the definition of the hyperbolic model in, e.g., Eqs. 1-3? Are the hidden degrees \mathbf{h} power-law

distributed according to γ , and the node coordinated distributed uniformly at random over the D-dimensional sphere? The description of synthetic network generation in Sec. V could be made clearer.

Reply: As the referee notes, synthetic networks were generated with the D model. In the new version of the manuscript, the model is described at the beginning of the results section.

The distribution of hidden degrees can be of any form. For the experiments in Fig. 2 and Fig. 3, we used power-law hidden degree distribution of the form $\rho(\kappa) = (\gamma - 1)\kappa^{\gamma-2}$ with a minimum value of the hidden degree that depends on the target average degree $\kappa > \kappa_0 = (\gamma - 2)/(\gamma - 1)$ (κ) and different values of the characteristic exponent $2 < \gamma < 3$. For every node i in the simulated network, we generated the hidden degree κ_i as a random value from this distribution. The position in the D-dimensional sphere of radius P representing the similarity space was chosen uniformly at random; each node was assigned a vector $v_i \in \mathbb{R}^{D+1}$, where initially a vector contained $\Delta+1$ standard normally distributed entries which was later normalized to lie on the sphere, with $\|v_i\| = P$, where P is given in Eq. (2) of the manuscript. The hidden degrees and the coordinates in the D-dimensional similarity space are used in Eq.(1) to calculate the probability of connection, with the selected value of β and the value of α in Eq. (3) depending on the target average degree (κ), β , and Δ .

Action taken: We added a section to the Methods to clarify the description of how synthetic networks were produced.

Comment 3.6:

- Sec. V C and Fig. 4: Synthetic networks with modular structure: How much community overlap is there as a function of the polar angle parameter? Does overlap affect the ability of the embedding method to recover the geometric concentration of communities?

Reply: This is an interesting question. To generate the synthetic communities, we defined spherical caps with the apices evenly distributed on the surface of the Δ -sphere and polar angle $\Delta\theta_T = 0.7$. Finally, nodes in each community were distributed homogeneously within the corresponding cap. In τ_1 and τ_2 networks with community structure, there is no overlap up to $\Delta\theta_T = \pi/4$. If we progressively increase $\Delta\theta_T$, the overlap grows continuously and faster for τ_2 network (see Figure 1 and Section IX.A in SI).

Figure 1: Overlap of communities as the function of $\Delta\theta_T$ in the synthetic networks. The results are averaged over 100 realizations.

We also carried out experiments with the larger value of $LO = 1.0$, thus accounting for the community overlap. Results of community concentration are shown in Figure 2 in this response (and Figure S19). We can observe similar behavior as in Figure 3 in the revised version of the manuscript. The embedding in a higher dimension is required to uncover the community structure generated in the higher dimension.

Figure 2: The community concentration cC for the (a) 1 model with 4 communities and (b) 2 model with 6 communities embedded in different dimensions when communities were overlapping, i.e., with $LO = 1.0$. The following parameters were used to generate the networks: $\beta = 1.5D$, $\gamma = 2.7$, $N = 2000$. Results are averaged over 10 realizations.

Overlap, thus, affect the ability of the embedding method to recover the geometric concentration of communities, but the value that we selected for the reported experiments in the manuscript remains unaltered.

Action taken: We added the study of community overlap as a function of LO to SI and a sentence summarizing the results to main text. Also, we included in SI the results of community concentration when we observe partial overlapping of communities, i.e., for $LO = 1.0$.

Comment 3.7:

MINOR COMMENTS

- Introduction, paragraph 2: ... directed network(s)... is there a missing 's'?
- After Eq. 8: Is $n_{i,i}$ actually $n_{i,i}$?

Reply and action taken: Thank you for pointing this out. Mistakes corrected.

REVIEWER COMMENTS

Reviewer #1 (Remarks to the Author):

I have been through the new version of the manuscript and the author's responses in detail, and I agree that the manuscript is improved from the previous version, including but not only because it is now in the correct format that I can assess its readability by a broader audience. However, I am sorry to say that I remain unconvinced that this paper rises to the level of being sufficiently "important" and "of significance" to need to be in the journal. Both I and another referee voiced a very similar variation of this same point, but I am not compelled by the responses to these questions.

Indeed, I disagree with some of the response made there by analogy to the role of dimensionality in the natural sciences; the better analogy I believe is to multidimensional scaling of point cloud data, where there is no one right answer for what low dimension or what kind of space to embed the data in but instead the possibility of multiple different good answers depending on the purpose of the dimensionality reduction being carried out. But this is a side disagreement independent of the problem to me that this revised manuscript still doesn't in my estimation sufficiently demonstrate that it meets the standards of importance and significance to be in the journal.

I also have some difficulty with the additional emphasis on community detection. There are many ways to do community detection in networks. Some reviews appear to be referenced but nothing is provided here about how the communities obtained here are similar to or different from what one might find using any of the many other standard methods for community detection that are available. Is this method of community detection somehow better compared to other state-of-the-art methods? Or is it merely more compellingly connected to these low dimensional representations here?

I appreciate the additional work that the authors put into this to rewrite it for the style of the journal and for adding more information about the real-world data. But I remain unconvinced that it rises to the demands of the journal. I strongly suggest resubmission to another venue that is either more specialized to these kinds of questions in the network science literature, or alternatively to a broader-interest journal like Scientific Reports that does not have the same editorial criteria as the present journal.

Reviewer #2 (Remarks to the Author):

The authors addressed all my comments. As I already stated in the previous round of review, this is a good paper and I support its publication in the journal.

Reviewer #3 (Remarks to the Author):

I'd like to thank the authors for a thorough revision of their manuscript, particularly the restructuring of the main text, and the rewritten/extended Introduction and Discussion sections. The new version makes a better job of arguing for the significance of the work in light of the recent Ref. [2] by the authors, stressing that the D-Mercator methodology allows us not only to infer the optimal dimension of a real network in a hyperbolic embedding, but also the distances and relative positions between pairs of nodes, which is useful for applications such as navigability and community detection. I also appreciate the extended and clarified description of the embedding method and the discussion of potential challenges in reaching global optima. The added analysis of the effect of community overlap on embedding efficiency is also interesting, and I'd be looking forward to further exploration of the limits of this methodology when applied to real networks with heavily overlapping community structure, as might be the case in e.g. social networks (in future research, naturally). Overall, I think the revised version of the manuscript warrants the broad readership of Nature Communications and successfully argues for its significance in the ongoing field of network geometry.

Reviewer #1

Comment 1.0.:

I have been through the new version of the manuscript and the author's responses in detail, and I agree that the manuscript is improved from the previous version, including but not only because it is now in the correct format that I can assess its readability by a broader audience. However, I am sorry to say that I remain unconvinced that this paper rises to the level of being sufficiently "important" and "of significance" to need to be in the journal. Both I and another referee voiced a very similar variation of this same point, but I am not compelled by the responses to these questions.

Reply: We thank the referee for their time, for their evaluation of our paper, and for recognizing the improvements over the previous version of the manuscript. We want to stress that the two other referees found the revised version of our manuscript suitable for publication in Nature Communications.

After this resubmission, we hope that the referee will recognize the value of our work and deem it suitable for publication in Nature Communications.

Comment 1.1.: Indeed, I disagree with some of the response made there by analogy to the role of dimensionality in the natural sciences; the better analogy I believe is to multidimensional scaling of point cloud data, where there is no one right answer for what low dimension or what kind of space to embed the data in but instead the possibility of multiple different good answers depending on the purpose of the dimensionality reduction being carried out. But this is a side disagreement independent of the problem to me that this revised manuscript still doesn't in my estimation sufficiently demonstrate that it meets the standards of importance and significance to be in the journal.

Reply: We agree that the problem of dimensions—identifying the dimensionality of the relevant space associated with a given phenomenon—is particularly challenging in complex systems and complex networks, and "the right answer" might not exist. However, there are answers which are more correct or more efficient than others, and definitely there are answers to this problem that require a much lower dimensionality as compared to other approaches.

In our case, we are concerned with complex relational datasets, and the purpose of our approach is to provide a model-based geometric description of the connectivity structure of networks that is consistent with their complex features. From a practical point of view, our approach can be seen as a dimensional reduction technique for complex relational datasets. But we are not proposing one more method for the dimensional reduction of point cloud data; we are proposing the multidimensional version of the best model-based embedding method currently available to explain the connectivity structure of complex networks.

The choice of hyperbolic rather than Euclidean geometry to depict the geometry of complex relational datasets and networks is crucial and essential. Most real complex networks share the same typical properties, such as heterogeneous degree distribution, small-world property, high clustering, or hierarchical structure. The hyperbolic representation can naturally capture and explain these attributes simulta-

neously due to the exponential expansion of space in hyperbolic geometry, without requiring a very large number of dimensions. The low dimensionality required for the hyperbolic representation offers convenient advantages over representations in Euclidean spaces that require much higher dimensions.

Let us finally mention that Multidimensional Scaling (MDS) is a method to find a lower dimensional representation of the data by displaying a dissimilarity matrix visually while preserving the pairwise distances as much as possible. The choice of the right dimension to represent the data is essential to avoid either underfitting or overfitting. Our multidimensional embedding method not only provides a visual representation of a real network but also enables us to use the obtained embedding, i.e., inferred parameters of the model, to generate synthetic network replicas with the same intrinsic properties as the analyzed network. Hence, we can revert the process, which is impossible using MDS. Moreover, the choice of the best dimension is supported by the features of the ensemble replicas—see Nat. Commun. 13, 6096 (2022) for details—and are therefore chosen based on several criteria, thereby not relying on one specific task.

Comment 1.2.: I also have some difficulty with the additional emphasis on community detection. There are many ways to do community detection in networks. Some reviews appear to be referenced but nothing is provided here about how the communities obtained here are similar to or different from what one might find using any of the many other standard methods for community detection that are available. Is this method of community detection somehow better compared to other state-of-the-art methods? Or is it merely more compellingly connected to these low dimensional representations here?

Reply: Notice that the focus of our work is not on community detection. We used greedy routing navigation and community detection as examples of tasks that can benefit from our multidimensional embeddings. The same for many other features or processes that are helped or sustained by the multidimensional geometric approach. For instance, we recently explored the emergence of geometric Turing patterns in dimension $D = 1$ [5]. Interestingly, we were able to detect such patterns in several real complex networks. However, if the optimal dimension for a given system is higher than $D = 1$, the potential Turing patterns will be distorted enormously in a one-dimensional representation. At present, we are working towards extending our approach to higher dimensions and, thus, it becomes critical to be able to infer the most appropriate dimension and the corresponding embedding. Other applications, such as geometric renormalization and link prediction, will definitely benefit from the multidimensional representation of complex networks.

Concerning communities, we agree that further research is required in the future and that there are many ways to do community detection in networks. The advantage of our geometric approach is that it opens the door to add the application of community detection methods based on geometry to the application of methods based on topology, and maybe new techniques merging the geometric and topological approaches will come in the future. To illustrate the community problem in our manuscript, we used one of the simplest geometric methods, i.e., the agglomerative clustering algorithm, which is a method that merges together points close to each other based on a predefined distance metric. However, the embedding obtained from our method would have allowed us to use many different clustering algorithms.

In any case, we extended our work by including a comparison between four state-of-the-art topological-based methods for community detection and the geometric agglomerative clustering algorithm used on the obtained embeddings in the best dimension. Table 1 shows the Normalized Mutual Information between the predicted communities and the metadata labels, whereas Table 2 shows the obtained values of modularity. Despite the simplicity of the agglomerative clustering method used, overall, the geometric community detection method exhibits comparable –and even better in some cases– outcomes

regarding modularity and NMI as compared to state-of-the-art community detection methods based on topology alone. It is also worth noting that while the Louvain algorithm performs well in maximizing the modularity, the number of clusters that the method finds is usually much higher than we observe in metadata. On the other hand, the maximum modularity is achieved with a similar number of clusters as in the metadata by utilizing our geometric-based approach (see Table 4). This means that the coordinates in hyperbolic space inferred by our method in the right dimension are meaningful, in the sense that they encode not only the local topology of the original network (who is likelier to be connected with whom), but also encode its mesoscopic organisation (groups or communities). Of course, our geometric community detection algorithm is rather simple and could be improved on many fronts, notably by designing or adopting a more appropriate algorithm to cluster embedded points in higher dimensions; however, this goes beyond the scope of our paper. Nevertheless, we believe that our conclusions open the possibility to extend clustering algorithms based on spatial ordering, in particular the exact clustering algorithm [6], to spaces of more than 1 dimension.

In Table 3, we report the overlap between the clusters obtained from the geometric and the topological-based methods. Whereas in some analyzed networks, the overlap achieves high values, there are cases where the geometric-based method groups nodes differently compared to the topological-based alternatives.

Finally, we would like to stress the major advantage of having a good embedding vs a partition of the network in different communities. In the later case, the only we can state is that two nodes in the same community are somehow “close” and two nodes in different communities are “far”. In the former, with the embedding in the right dimension, we can define a $N \times N$ matrix quantifying the closeness between any pair of nodes in the system. This represents a major improvement of the description of the system, which allows not only to detect communities but also has many other applications, for instance, navigability and greedy routing, geometric renormalization and scale up and down network replicas, Turing patterns, link prediction, etc.

Table 1: Comparison of the community detection performance in terms of **Normalized Mutual Information** (NMI) between the predicted communities and the metadata labels for the agglomerative clustering algorithm based on the embeddings in the best dimension and the topological based methods: GMM (greedy modularity maximization) [1], Louvain method [2], Infomap [3] and LPA (Label Propagation Algorithm) [4]. For the agglomerative clustering algorithm we report two cases: (i) where the number of clusters is obtained from the metadata and (ii) when the number of clusters is determined by the maximum modularity. The highest value is shown in **blue** and the second highest in **orange**.

	agglomerative clustering	GMM	Louvain	Infomap	LPA
Add-health	0.4854 / 0.4717	0.322	0.374	0.4141	0.3608
FAO-apples	0.391 / 0.3218	0.3869	0.5424	0.1432	0.024
C. elegans	0.4938 / 0.4938	0.4146	0.4653	0.4446	0.4305
OpenFlights	0.6078 / 0.6218	0.5957	0.7294	0.6168	0.5599
Foxglove	0.3321 / 0.3382	0.044	0.3437	0.2221	0.3130
Polbooks	0.5666 / 0.5299	0.5308	0.5901	0.5288	0.4383

Table 2: Comparison of the community detection performance in terms of **modularity (Q)** between the agglomerative clustering algorithm based on the embeddings in the best dimension and the topological based methods: GMM (greedy modularity maximization) [1], Louvain method [2], Infomap [3] and LPA (Label Propagation Algorithm) [4]. For the agglomerative clustering algorithm we report two cases: (i) where the number of clusters is obtained from the metadata and (ii) when the number of clusters is determined by the maximum modularity. The highest value is shown in **blue** and the second highest in **orange**.

	agglomerative clustering	GMM	Louvain	Infomap	LPA
Add-health	0.5788 / 0.5796	0.5413	0.6138	0.4182	0.5015
FAO-apples	0.1555 / 0.2029	0.2334	0.2589	0.0146	0.0016
C. elegans	0.513 / 0.513	0.4877	0.5284	0.5245	0.4604
OpenFlights	0.5425 / 0.5659	0.6	0.6582	0.5622	0.6129
Foxglove	0.6952 / 0.6954	0.5983	0.7484	0.6506	0.5732
Polbooks	0.4954 / 0.5123	0.5019	0.527	0.5259	0.4817

Table 3: The overlap between the communities obtained by applying the agglomerative clustering algorithm based on the embeddings in the best dimension and four topological based methods in terms of Normalized Mutual Information (NMI). We report two cases: (i) where the number of clusters is obtained from the metadata and (ii) when the number of clusters is determined by the maximum modularity.

	GMM	Louvain	Infomap	LPA
Add-health	0.4129 / 0.411	0.5346 / 0.5543	0.4496 / 0.4104	0.4162 / 0.4543
FAO-apples	0.3638 / 0.3940	0.516 / 0.4664	0.1295 / 0.1298	0.029 / 0.0254
C. elegans	0.6235 / 0.6235	0.6258 / 0.6258	0.6443 / 0.6443	0.6179 / 0.6179
OpenFlights	0.5999 / 0.6019	0.6363 / 0.6485	0.7227 / 0.7698	0.5451 / 0.5519
Foxglove	0.3626 / 0.3927	0.5478 / 0.5943	0.4684 / 0.4852	0.4913 / 0.5146
Polbooks	0.8616 / 0.7732	0.8107 / 0.8502	0.8225 / 0.8430	0.6649 / 0.7184

Table 4: The number of clusters found by community detection methods. For the agglomerative clustering algorithm we report two cases: (i) where the number of clusters is obtained from the metadata and (ii) when the number of clusters is determined by the maximum modularity.

	agglomerative clustering	GMM	Louvain	Infomap	LPA
Add-health	6 / 9	12	10	2	184
FAO-apples	6 / 3	5	4	5	2
C. elegans	5 / 5	5	6	10	4
OpenFlights	6 / 5	41	25	4	4
Foxglove	8 / 10	4	15	7	145
Polbooks	3 / 4	4	5	5	8

[1] Clauset, A., Newman, M. E., & Moore, C. (2004). Finding community structure in very large networks. *Physical review E*, **70(6)**, 066111.

[2] Blondel, V. D., Guillaume, J. L., Lambiotte, R., & Lefebvre, E. (2008). Fast unfolding of communities in large networks. *Journal of statistical mechanics: theory and experiment*, **2008(10)**, P10008.

[3] Rosvall, M., & Bergstrom, C. T. (2008). Maps of random walks on complex networks reveal community structure. *Proceedings of the national academy of sciences*, **105(4)**, 1118-1123.

[4] Raghavan, U. N., Albert, R., & Kumara, S. (2007). Near linear time algorithm to detect community structures in large-scale networks. *Physical review E*, **76(3)**, 036106.

[5] Jasper van der Kolk, Guillermo García-Pérez, Nikos E. Kouvaris, M. Ángeles Serrano, and Marián Boguñá. (2023). Emergence of geometric Turing patterns in complex networks. *Physical Review X* 13 021038 (2023)

[6] Alice Patania, Antoine Allard, & Jean-Gabriel Young. (2023). Exact and rapid linear clustering of networks with dynamic programming. *Proceedings of the Royal Society A*, 479, 20230159.

Action taken: We included the comparison between the geometric and topological-based community detection method in the SI and added the corresponding paragraph in the main text.

Comment 1.3.: I appreciate the additional work that the authors put into this to rewrite it for the style of the journal and for adding more information about the real-world data. But I remain unconvinced that it rises to the demands of the journal. I strongly suggest resubmission to another venue that is either more specialized to these kinds of questions in the network science literature, or alternatively to a broader-interest journal like *Scientific Reports* that does not have the same editorial criteria as the present journal.

Reply: We hope that the new version of the manuscript is more clear and convincing.

Reviewer #2

Comment 2.0: The authors addressed all my comments. As I already stated in the previous round of review, this is a good paper and I support its publication in the journal.

Reply: We thank the reviewer for their time and effort in reading our paper and for their report. We are delighted to learn that the reviewer considers our paper well-written and deems it suitable for publication in *Nature Communications*.

Reviewer #3

Comment 3.0: I'd like to thank the authors for a thorough revision of their manuscript, particularly the restructuring of the main text, and the rewritten/extended Introduction and Discussion sections. The new version makes a better job of arguing for the significance of the work in light of the recent Ref. [2] by the authors, stressing that the D-Mercator methodology allows us not only to infer the optimal dimension of a real network in a hyperbolic embedding, but also the distances and relative positions between pairs of nodes, which is useful for applications such as navigability and community detection. I also appreciate the extended and clarified description of the embedding method and the discussion of potential challenges in reaching global optima. The added analysis of the effect of community overlap on embedding efficiency is also interesting, and I'd be looking forward to further exploration of the limits of this methodology when applied to real networks with heavily overlapping community structure, as might be the case in e.g. social networks (in future research, naturally). Overall, I think the revised version of the manuscript warrants the broad readership of *Nature Communications* and successfully argues for its significance in the ongoing field of network geometry.

Reply: We thank the reviewer for their time and effort in reading our paper and for their report. We are delighted to learn that the reviewer considers our paper well-written and deems it suitable for publication in *Nature Communications*.

REVIEWERS' COMMENTS

Reviewer #1 (Remarks to the Author):

I have been over the new version in detail. I appreciate the additional effort done by the authors to further expand the details of their tie-in to community detection. I also appreciate the additional comments added to the main text about "[o]ther lines of research for future work" (though I will say, frankly, it's a little disappointing that 3 of the 4 citations there are self-cites - surely there are others you could have cited there? - if you want to change or add cites in a final revision I'll leave it to you)

Given this additional level of added detail, and since I am now the outlier among the referees, I will capitulate and recommend this version be accepted.

Reviewer #1

Comment 1.0.:

I have been over the new version in detail. I appreciate the additional effort done by the authors to further expand the details of their tie-in to community detection. I also appreciate the additional comments added to the main text about "[o]ther lines of research for future work" (though I will say, frankly, it's a little disappointing that 3 of the 4 citations there are self-cites - surely there are others you could have cited there? - if you want to change or add cites in a final revision I'll leave it to you)

Given this additional level of added detail, and since I am now the outlier among the referees, I will capitulate and recommend this version be accepted.

Reply and action taken: We thank the referee for their time and for their evaluation of our paper, and for the comments that have certainly helped to improve the manuscript. We have included additional lines of future work in the paper, along with proper citations.